# Impact of palladium/palladium hydride conversion on electrochemical CO$_2$ reduction via in-situ transmission electron microscopy and diffraction

Ahmed M. Abdellah[1], Fatma Ismail[1], Oliver W. Siig [2], Jie Yang[3], Carmen M. Andrei[4], Liza-Anastasia DiCecco[5], Amirhossein Rakhsha[1], Kholoud E. Salem[1], Kathryn Grandfield [3,5], Nabil Bassim [3,4], Robert Black[6], Georg Kastlunger [2] ✉, Leyla Soleymani [5,7] & Drew Higgins [1,4] ✉

Electrochemical conversion of CO$_2$ offers a sustainable route for producing fuels and chemicals. Pd-based catalysts are effective for converting CO$_2$ into formate at low overpotentials and CO/H$_2$ at high overpotentials, while undergoing poorly understood morphology and phase structure transformations under reaction conditions that impact performance. Herein, in-situ liquid-phase transmission electron microscopy and select area diffraction measurements are applied to track the morphology and Pd/PdH$_x$ phase interconversion under reaction conditions as a function of electrode potential. These studies identify the degradation mechanisms, including poisoning and physical structure changes, occurring in PdH$_x$/Pd electrodes. Constant potential density functional theory calculations are used to probe the reaction mechanisms occurring on the PdH$_x$ structures observed under reaction conditions. Microkinetic modeling reveals that the intercalation of *H into Pd is essential for formate production. However, the change in electrochemical CO$_2$ conversion selectivity away from formate and towards CO/H$_2$ at increasing overpotentials is due to electrode potential dependent changes in the reaction energetics and not a consequence of morphology or phase structure changes.

Electrochemical conversion of CO$_2$ using renewable electricity is envisaged as an integral component of a future sustainable energy economy by providing an avenue for producing carbon-based fuels and chemicals from non-fossil fuel feedstocks. Nevertheless, electrochemical CO$_2$ conversion technologies require efficient, selective, and stable electrocatalysts for CO$_2$ reduction (CO$_2$R) reactions with these performance traits fundamentally dictated by the properties of the materials under reaction conditions. The activity and selectivity of catalysts for CO$_2$R and the competing hydrogen evolution reaction (HER) are known to depend on the binding energies between the active surface of the materials and adsorbed reaction intermediates, for example, *CO or *H, respectively, where * denotes an adsorbed

[1]Department of Chemical Engineering, McMaster University, Hamilton, ON, Canada. [2]CatTheory, Department of Physics, Technical University of Denmark, Kongens Lyngby, Denmark. [3]Department of Materials Science and Engineering, McMaster University, Hamilton, ON, Canada. [4]Canadian Centre for Electron Microscopy, McMaster University, Hamilton, Canada. [5]School of Biomedical Engineering, McMaster University, Hamilton, Canada. [6]National Research Council of Canada, Energy, Mining, and Environment Research Centre, Mississauga, ON, Canada. [7]Department of Engineering Physics, McMaster University, Hamilton, Canada. ✉e-mail: geokast@dtu.dk; higgid2@mcmaster.ca

species[1]. The binding energies between adsorbed species and the catalyst surface are dictated by the nature of the catalytically active site structure(s) present in the catalyst materials and can be modulated by tuning the catalyst properties by strategies such as alloying[2–5], surface modification or the exposure to different surface facets[6–8].

Palladium-based electrocatalysts provide the lowest known overpotential for $CO_2R$ among all reported catalysts[9,10]. Particularly, Pd exhibits high selectivity (> 90%) for reducing $CO_2$ into formate at low overpotentials (< 200 mV)[9]. At higher overpotentials, Pd catalysts become selective towards the formation of $H_2$ and CO, including Faradaic efficiencies towards CO above 90% recorded at overpotentials of ca. 500 mV[10]. Under electrochemical $CO_2R$ conditions, Pd undergoes transformation into Pd-hydride ($PdH_x$) phases[9–15]. Based on the stoichiometric ratio of hydrogen (x) in $PdH_x$, α- and β-phase $PdH_x$ are formed in the range of $0 < x \leq 0.03$ and $x \geq 0.58$, respectively, while α- and β-phase $PdH_x$ coexist in the $0.03 < x < 0.58$ region[16,17]. These phase transformations cause the surface structure, electronic properties, and lattice spacing of the $PdH_x$ catalyst to vary, thus affecting catalytic activity and selectivity[18–22]. Furthermore, the (electro)chemical environment experienced under $CO_2R$ conditions causes changes to the structure and properties of the electrocatalyst[7,9,23,24]. Observing these marked changes in the structure and properties of the materials under reaction conditions alongside correlating the results with measured catalytic properties can provide crucial mechanistic insight into catalytic activity and stability.

Various techniques have been developed to investigate $Pd/PdH_x$ phase transformations and/or quantify the resulting H: Pd ratios. Using deuterium (D) instead of hydrogen, D: Pd ratios have been quantified by electrolytically forming $PdD_x$ structures and then liberating $D_2$ by heating and measuring the amount of gas released[25,26]. Such techniques are not amenable to $Pd/PdH_x$ catalysts under $CO_2$ reduction conditions, which require applying external electrochemical potentials in the presence of liquid electrolytes. In-situ (operando) measurements including X-ray diffraction (XRD) and extended X-ray absorption fine structure (EXAFS) have been employed to identify the lattice parameters and interatomic distances of $Pd/PdH_x$ phases, respectively[27–31]. Landers et al.[32] utilized in-situ synchrotron XRD and coulometry measurements to understand intercalation/deintercalation processes for hydrogen in palladium, enabling the determination of the electrode potentials where α- and $β-PdH_x$ phases were formed[32]. Gao et al.[10] utilized in-situ XAS to demonstrate the coexistence of α- and β-phases at potentials above −0.2 V (vs. the reversible hydrogen electrode, RHE), which promoted electrochemical $CO_2R$ into formate via an HCOO* intermediate. At potentials below −0.5 V vs. RHE, the formation of $β-PdH_x$ was observed and claimed to promote the formation of CO via a COOH* intermediate[10]. While these in-situ synchrotron-based techniques enable monitoring of phase structure transformations in the active $PdH_x/Pd$ materials as a function of electrode potential, they do not provide the opportunity to observe morphological changes in the catalyst particles under $CO_2$ reduction conditions that have a direct implication on catalytic activity and stability.

In-situ liquid-phase (scanning) transmission electron microscopy (LP-(S)TEM) provides the opportunity to observe morphological/ compositional changes of catalysts under electrochemical conditions[23,33–36], while also enabling analyses of the phase structure(s) by employing select area diffraction (SAD) or fast-Fourier transform (FFT) characterization. With appropriate instrumentation, these measurements can be conducted at electron microscopy facilities, which for many researchers are more readily available and accessible than specialized research facilities, such as synchrotrons. To this end, using in-situ LP-(S)TEM, advanced insight into catalyst properties with spatial resolution under reaction conditions can be achieved by correlating morphological imaging with analytical techniques such as SAD and energy dispersive X-ray (EDX) analysis. A previously in-situ LP-(S)TEM

study monitored the morphological evolution of Pd particles under electrochemical conditions, including during the electrodeposition of Pd particles[37] or their morphological evolution under potential cycling[38], but this work was not done in the context of electrocatalysis and the formation of $PdH_x$ phases was not probed. The formation of $PdH_x$ phases has been imaged previously by in-situ TEM, however, these studies were conducted using either an in-situ environmental gas cell[39–42] or via in-situ LP-(S)TEM measurements[43]. These measurements were done in the absence of electrode potential and are therefore not pertinent to electrochemical $CO_2R$ investigations.

Despite advancements in in-situ LP-(S)TEM capabilities, detailed investigations of morphological changes in Pd-based catalysts under $CO_2R$ conditions have not been investigated. Furthermore, implementation of in-situ SAD measurements under electrochemical conditions during LP-(S)TEM workflows has never been reported, yet provides the opportunity to simultaneously measure and track phase structure transformations in catalysts under reaction conditions. Herein, in-situ LP-(S)TEM measurements on electro-deposited Pd/ $PdH_x$ catalysts were employed to track morphological changes under electrochemical $CO_2R$ conditions alongside LP-TEM/SAD patterns collected to probe the interconversion between metallic Pd and $PdH_x$ phases. Distinct morphological changes occurring in the catalyst structures under electrochemical $CO_2R$ conditions were observed alongside a phase transformation from metallic Pd to $PdH_x$ at electrochemical $CO_2R$ relevant potentials. Increasing lattice expansion due to increased absorption of H atoms occurred at more negative electrode potentials, seemingly giving rise to dramatic $CO_2R$ selectivity changes from nearly exclusive production of formate at −0.2 V vs RHE towards the production of CO and $H_2$ at −0.5 V vs RHE. The impact of the observed transition from Pd to $PdH_x$ was explored by density functional theory (DFT) calculations. Micro-kinetic analyses based on the latter, indicate that the production of formate is reliant on the presence of surface bound hydrogen, whose abundance increases with cathodic overpotential. However, the $CO_2R$ selectivity shift results from the varying responses in terms of the reaction energetics to the applied electrode potential of the formate and CO reaction pathways, with the latter benefitting more from increased cathodic overpotentials and not due to the phase structure transformations. Ultimately, in-situ LP-(S)TEM imaging coupled with SAD analysis has been demonstrated as an effective technique for gaining fundamental insight into the $PdH_x/Pd$ conversion (hydrogen uptake) process that has been of marked interest to the materials science community[40,43–51] with relevance in the fields of hydrogen storage, sensors and catalysis[22,45,52,53]. Particularly, $PdH_x/Pd$ catalysts are characterized under electrochemical $CO_2R$ conditions for simultaneously observing morphology and $PdH_x/Pd$ phase structure changes to identify catalytically active material degradation pathways, alongside these in-situ measurements coupled with electro-catalytic activity/selectivity evaluation and computational analysis to provide new mechanistic insight into Pd-based catalysts for the electrochemical $CO_2R$.

## Results
### Electrode preparation
Figure 1 shows schematics of the two $CO_2$ electrolysis configurations used in this work. The first is the Protochips Poseidon Select in-situ LP-TEM electrochemical liquid cell microchip reactor (Fig. 1a) employing a glassy carbon working electrode decorated with Pd particles for in-situ characterization under $CO_2R$ conditions. The second is a two-compartment cell consisting of a large-format working electrode (Fig. 1b) decorated with Pd particles to measure catalytic activity and selectivity towards $CO_2R$. Pd particles were deposited on the microchip glassy carbon working electrode (Fig. 1c, and Supplementary Fig. 1) by electrodeposition after assembling the in-situ electrochemical TEM liquid reactor, and the same electrodeposition procedure was used for preparing large-format electrodes (Fig. 1d).

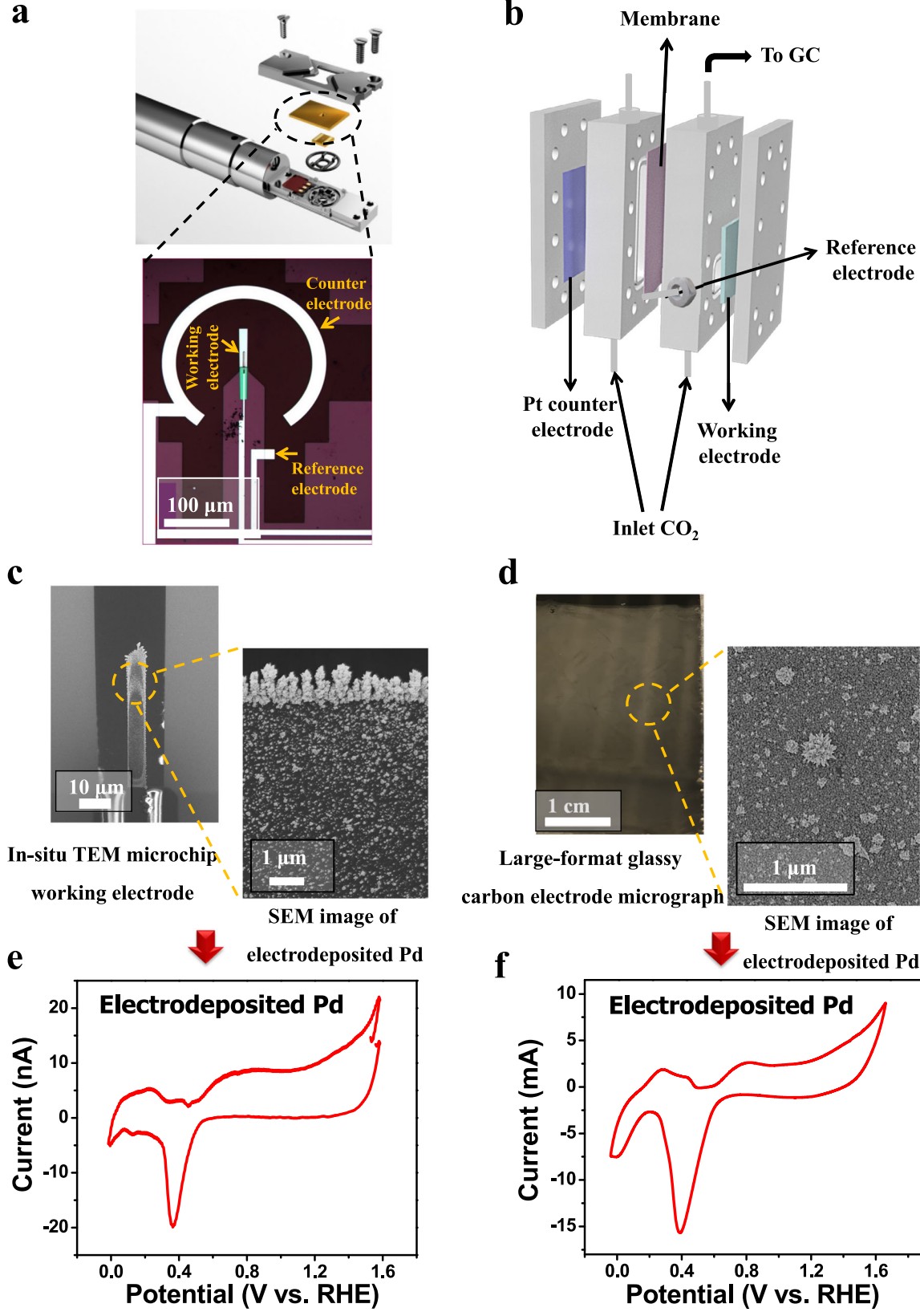

Electro-deposition of Pd particles on both the microchip (Fig. 1c) and the large format electrode (Fig. 1d) was observed by SEM imaging. Cyclic voltammetry measurements were collected in $N_2$ saturated 0.1 M KHCO$_3$ for the Pd particle decorated microchip glassy carbon working electrode (Fig. 1e) and the Pd decorated large-format electrodes (Fig. 1f), demonstrating similar features that suggest similar local chemical environments at the Pd-decorated working electrode in the two different electrochemical reactors. The redox characteristics observed via cyclic voltammetry were characteristic of Pd-based catalysts[9]. By comparing the cyclic voltammetry features of Pd, the Pt reference electrode employed in the microchip reactor was calibrated to the RHE scale, whereby 0.76 V vs. Pt corresponds to 0 V vs. RHE

**Fig. 1 | Schematics of the two CO₂ electrolysis cells utilized in this work.**
**a** Protochips Poseidon in-situ LP-(S)TEM holder consisting of a Pd decorated glassy carbon working electrode within a microchip electrochemical cell. **b** Two-compartment electrochemical cell consisting of a large-format Pd decorated glassy carbon working electrode for electrochemical CO₂R activity and selectivity measurements. **c** SEM images of the in-situ TEM microchip working electrode coated with electrodeposited Pd particles. **d** Micrograph of the large-format glassy carbon electrode and SEM image of the electrodeposited Pd particles. **e** Cyclic voltammetry measurements of electrodeposited Pd particles measured in the in-situ TEM microchip electrochemical cell. **f** Cyclic voltammetry measurements of electrodeposited Pd particles measured in the two-compartment cell using the large-format electrode. Note that all cyclic voltammetry measurements were collected in N₂-saturated 0.1 M KHCO₃ at a scan rate of 50 mV/s.

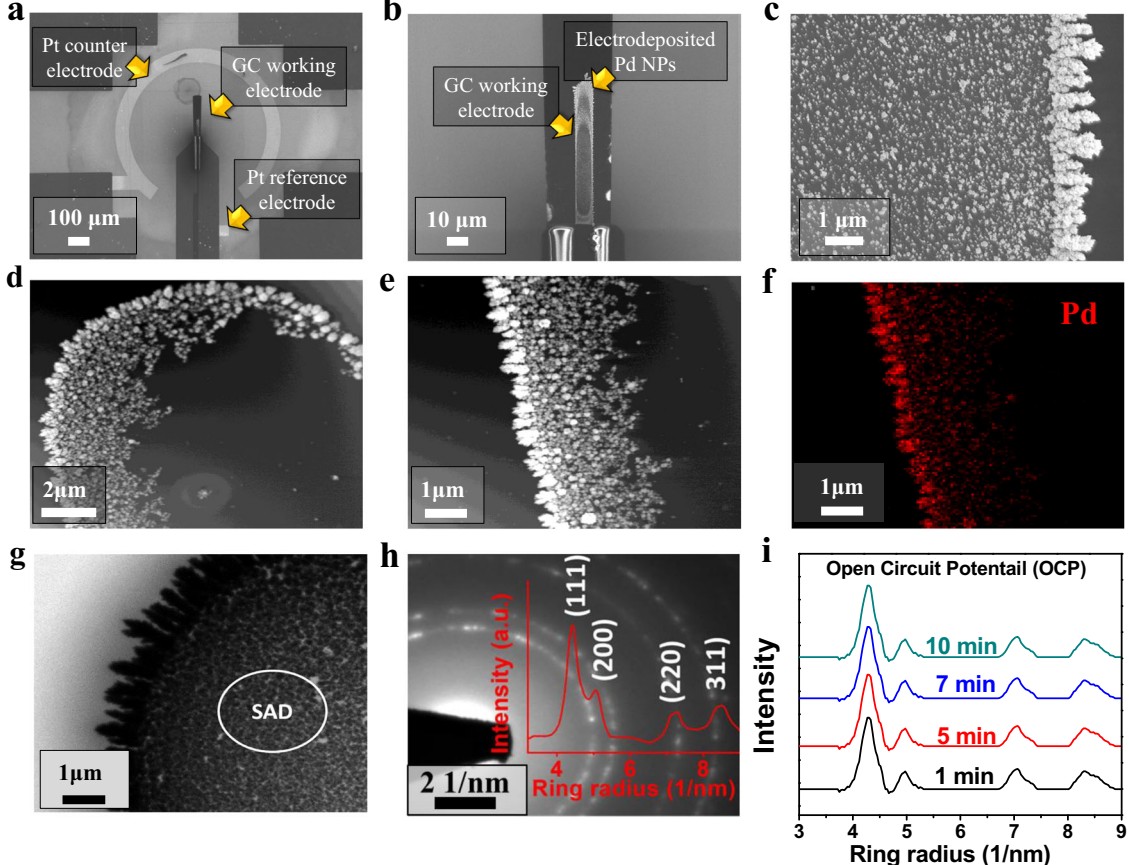

**Fig. 2 | Structural characterization of Pd catalysts deposited on the electrochemical in-situ LP-(S)TEM microchip. a** SEM image of the three electrode-configuration of the in-situ LP-(S)TEM microchip cell. **b**, **c** SEM of the working electrode showing electro-deposited Pd particles. **d**, **e** In-situ LP-HAADF-STEM images of the working electrode at various magnifications. **f** In-situ LP-HAADF-STEM-EDX mapping of the electrodeposited Pd particles on the working electrode. **g** In-situ LP-TEM imaging of Pd particles indicating the region where SAD patterns were measured. **h** In-situ LP-TEM/SAD pattern of Pd particles at open circuit potential and radial intensity profiles of the diffraction patterns using the in-situ electrochemical TEM liquid reactor. **i** Radial intensity profiles of the diffraction patterns at different beam irradiation times at a flow rate of 5 µL/min and a beam dose of 39.7 e⁻/nm² s. The acquisition time for the SAD patterns was set to be 1 s.

(Supplementary Fig. 2), in close agreement with previously reported values[23,33]. All subsequent potentials will be reported versus RHE.

SEM was employed to probe the morphology of electrodeposited Pd within the in-situ electrochemical TEM microchip reactor (Fig. 2a–c), revealing semi-spherical particles covering the electrode surface and dendrimer-type structures at the electrode edges. Figure 2d, e shows in-situ liquid phase high-angle annular dark-field scanning transmission electron microscopy (LP HAADF-STEM) images at various magnifications and Fig. 2f shows in-situ LP-HAADF-STEM/EDX mapping of the deposited Pd. A similar morphology was observed for the Pd electrodeposited on the large-format glassy carbon electrodes prepared for CO₂R activity and selectivity measurements (Supplementary Fig. 3).

**In-situ (S)TEM/SAD**
In-situ LP-TEM and SAD (LP-TEM/SAD) characterizations were conducted to investigate the morphology and phase structures of Pd

particles under electrochemical CO₂R conditions. In general, it is quite difficult to see clear diffraction rings in LP-TEM experiments due to the background generated by the diffuse scattering of electrons by the electrolyte. In this work, we improved the visibility of the diffraction patterns and increased their resolution by reducing the electrolyte thickness. The relative liquid thickness (t/λ) was reduced to (t/λ = 0.47 ~ 50 nm) as estimated by electron energy loss spectroscopy (EELS) analysis shown in Supplementary Fig. 4. The electrolyte thickness was reduced by implementing several approaches including utilizing a small E-chip without any spacer, with only a ~500 nm spacer present in the larger E-chip. Moreover, a cross-window configuration between the two SiNₓ windows was adopted, which reduced the electrolyte thickness near the edge as demonstrated previously[54]. Moreover, the formation of gas bubbles at the working electrode was another feature of electrocatalysis that was leveraged, whereby gas species like carbon monoxide and hydrogen are generated under applied electrochemical conditions. Formed gas bubbles can effectively purge most of the

electrolyte away from the $SiN_x$ window area, leaving behind a thin film of the liquid electrolyte that covers the surface of the E-chip to still provide ionic conductivity between all three electrodes while minimizing signal attenuation from the electrolyte as demonstrated previously[55,56].

Prior to the measurements, in-situ LP-TEM images (Fig. 2g) and in-situ LP-TEM/SAD patterns with radial intensity profiles (Fig. 2i) were collected at open circuit potential. During these measurements, we did not focus on the dendrite-like particles deposited at the edge of the working electrode as we feared they were not adequately adhered and could be electronically isolated from the working electrode surface and had a morphology that differed from the catalyst particles on the large-format electrodes used for $CO_2R$ activity and selectivity testing. Thus, we focused on catalyst particles within the interior of the working electrode as they were more representative of the catalyst particles for which $CO_2R$ activity and selectivity data was obtained. To ensure the diffraction patterns observed were from the Pd particles, in-situ LP-TEM/SAD patterns were also collected from an area of the electrode that did not contain any Pd particles (Supplementary Fig. 5a). No diffraction rings or spots were observed (Supplementary Fig. 5b), indicating that the glassy carbon working electrode, the $SiN_x$ windows, and the electrolyte did not contribute to the measurements. Repeated in-situ LP-TEM/SAD measurements were conducted on the Pd particles at open circuit potential to evaluate if the electron beam dose applied (39.7 electron/$nm^2$.frame, whereby each frame = 1 second) had any impact on the phase of the Pd particles. No phase transformations as a function of beam dose were observed, in agreement with a recent report investigating the impact of beam dose on Pd/$PdH_x$ interconversion, indicating no Pd to $PdH_x$ transformations occurred at beam doses as high as ca. 3900 electron/$nm^2$.sec[43]. In the present work no morphology changes in Pd were observed during LP-TEM measurements at open circuit potential, indicating the electron beam dose employed has limited effect on the induced morphology or phase structure changes.

In-situ LP-(S)TEM imaging of Pd particles in $CO_2$ saturated 0.1 M $KHCO_3$ as a function of electrode potential was conducted, with results illustrated in Fig. 3. HAADF-STEM images at 1.2 V vs RHE and after 27 seconds of applying an electrode potential of -0.2 V vs. RHE (extracted from Supplementary Movie 1) are shown in Fig. 3a, b, respectively. The corresponding in-situ LP-TEM/SAD patterns of the Pd particles are also shown for an electrode potential of 1.2 V vs RHE (Fig. 3c) and -0.2 V vs RHE (Fig. 3d). Going from 1.2 to -0.2 V vs RHE, the size of the Pd particles increased (Fig. 3b, with images from additional locations shown in Supplementary Fig. 6) and the radial distance of the SAD patterns noticeably decreased, suggesting an expansion of the Pd crystal lattice (Supplementary Movie 2). To analyze collected in-situ LP-TEM/SAD patterns, radial intensity profiles were extracted using the CrysTBox-ringGUI[57] with an error in d-spacing caused by the ellipticity to be within ±0.02 Å. The crystallographic information file (cif) acquired from the crystal structure database was used to index the patterns. Examples are shown in Fig. 3c, d, Supplementary Fig. 6, and Supplementary Fig. 7. This method was used to calculate the d-spacing of the Pd-based particles from in-situ LP-TEM/SAD patterns, which increased upon applying an electrode potential of -0.2 V vs RHE (Supplementary Movie 2). The observed lattice expansion is likely attributed to hydrogen absorption and intercalation into Pd, resulting in the formation of different $PdH_x$ phases.

To more closely probe the phase transformations occurring under electrochemical $CO_2R$ conditions, in-situ LP-TEM/SAD patterns of Pd-based catalysts were collected at several $CO_2R$-relevant electrode potentials. Before applying $CO_2R$-relevant potentials, a potential of 1.2 V vs. RHE (significantly more anodic than $CO_2R$ conditions) was applied for 60 s to ensure the Pd catalyst was in the same starting state before all in measurements as hysteresis in the Pd/$PdH_x$ conversion is a known phenomenon[32]. After 60 seconds at 1.2 V vs. RHE, the Pd

particles were in their metallic state (Supplementary Fig. 7) and the electrode potential was stepped to progressively more negative values (although always returning back to 1.2 V vs. RHE between each potential). The electrode potential profiles used in this work are depicted in Supplementary Fig. 9, and in-situ LP-TEM/SAD patterns collected at each measurement potential are shown in Fig. 3e. Peak locations from the in-situ LP-TEM/SAD patterns were used to calculate crystal lattice spacing values which are shown in Fig. 3f for the (111) diffraction plane.

In-situ LP-TEM/SAD measurements indicated crystal lattice expansion and compression based on the applied electrode potential. At 0.6, 0.8, and 1.2 V vs. RHE, Pd is in the metallic fcc form with Pd(111) d-spacings of 2.27 Å, 2.26 Å, and 2.25 Å, respectively, in alignment with the theoretical value of 2.25 Å obtained from the cif database using CrysTBox-ringGUI (Supplementary Fig. 7 and Supplementary Table 1)[57] shown as the dashed blue line in Fig. 3f. At 0.3 V vs. RHE, an increase in the d-spacing value to 2.31 Å was observed, likely due to the formation of the α-$PdH_x$ phase and consistent with the recent study in-situ XRD study by Landers et al.[32] In the present work, as the electrode potential was stepped more negatively to -0.1 V vs. RHE, a further lattice expansion to 2.34 Å was calculated from the diffraction pattern, likely attributed to the formation of a mixture of the α- and β-$PdH_x$ phases in agreement with previously reported DFT calculation[10]. The dashed red line in Fig. 3f represents the theoretical value for β-phase $PdH_x$, obtained from the cif database using CrysTBox-ringGUI; whereas Supplementary Fig. 8 and Supplementary Table 2 demonstrate the comparison of the theoretical and experimental values of $PdH_x$. At electrode potentials below -0.2 V vs. RHE, β-$PdH_x$ was found to be the predominant phase with further lattice expansion. Noted that all the experimental d-spacing calculations were estimated using the Crystal Box software with an accuracy ± 0.02 Å.

## Morphology Changes During $CO_2R$ Imaged by In-situ LP-TEM

Under electrochemical $CO_2R$ conditions, the morphology and behavior of Pd particles evolve over time[9,13]. To investigate these phenomena, in-situ LP-(S)TEM measurements were conducted on the $PdH_x$ particles at -0.2 V vs. RHE (Fig. 4a, Supplementary Fig. 10, Supplementary Movie 3 and Supplementary Movie 4). At an applied potential of -0.2 V vs. RHE, some $PdH_x$ particles were detached from the electrode surface after 5 s and found to migrate to another region of the electrode (Fig. 4a). For example, the particles in the region labeled P1 at t = 5 seconds migrated to the position labeled P2 after 16 seconds and beyond, demonstrating that detachment and aggregation of Pd-based particles occur under $CO_2R$ conditions. The average $PdH_x$ particle size was monitored in real-time by in-situ LP-TEM (Fig. 4a) at -0.2 V vs RHE and after all in-situ LP-(S)TEM measurements were conducted, ex-situ TEM (Fig. 4b and Supplementary Fig. 11), HAADF-STEM (Fig. 4c and Supplementary Fig. 12), SEM (Fig. 4d) and optical/SEM images (Supplementary Fig. 13) were conducted. Overall, an increase in Pd-based particle size from 80 ± 30 nm to 130 ± 30 nm was observed, indicating particle growth and agglomeration during electrochemical $CO_2R$. Additionally, the morphology of the Pd-based particles evolved into hollowed-out sponge-like porous structures that are most clearly depicted in the HAADF-STEM images in Supplementary Fig. 12 that provide contrast between the Pd atoms and void spaces. Post $CO_2R$ ex-situ characterization additionally revealed that Pd/$PdH_x$ particles detached from the electrode surface in various locations across the electrode (Supplementary Figs. 11-13). This detachment could be linked to mechanical stresses arising from the phase transitions between $PdH_x$ and Pd, which are accompanied by particle volume expansion/contraction and can induce deformation mechanisms[58]. This could also be impacted by changes to the surface chemistry of the carbon electrode under electrochemically reducing conditions, which may weaken the interactions between the Pd/$PdH_x$ particles and the carbon electrode, rendering the particles more prone

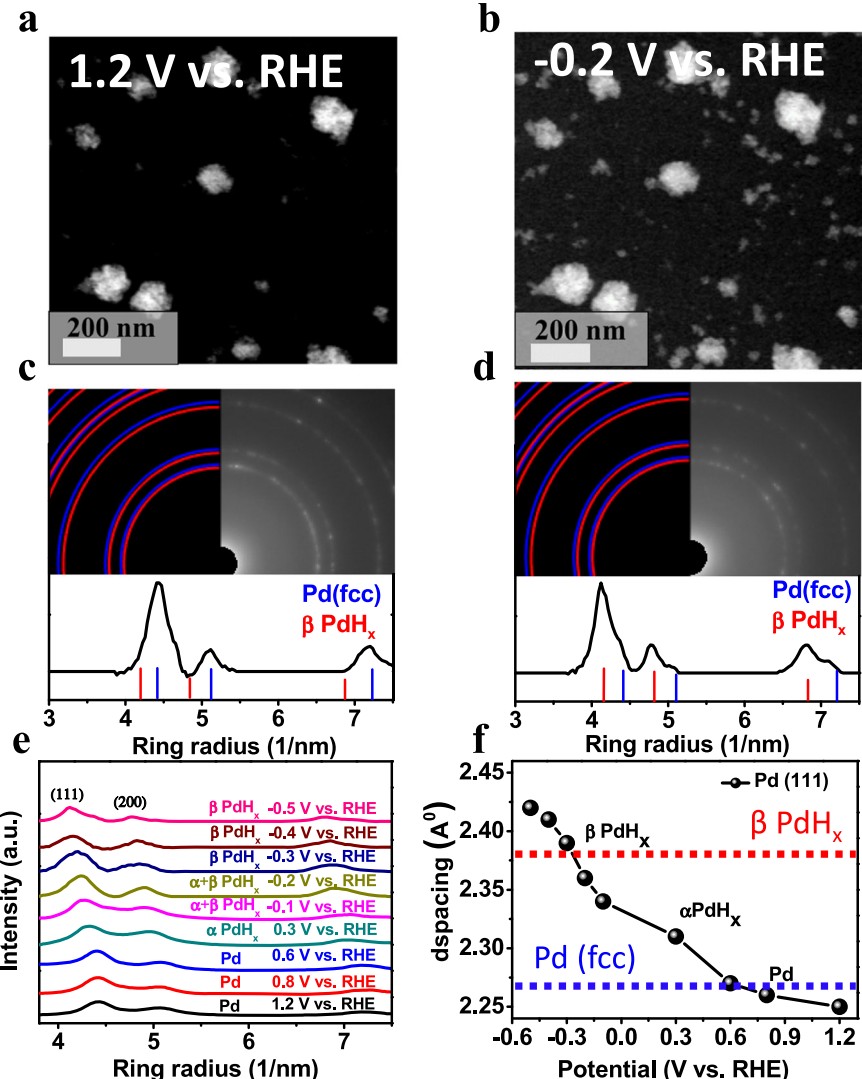

**Fig. 3 | In-situ LP-HAADF-STEM imaging and SAD patterns illustrate lattice expansion due to the phase transformation of metallic Pd to PdH$_x$ under electrochemical CO$_2$ reduction conditions. a, b** In-situ LP-HAADF-STEM snapshots extracted from Supplementary Movie 1 revealing enlargement of Pd particles under applied potential (1.2 V and −0.2 V vs. RHE, respectively) in CO$_2$ saturated 0.1 M KHCO$_3$ electrolyte. **c, d** In-situ LP-TEM-SAD patterns and corresponding radial intensity profiles revealing lattice expansion under applied electrode potentials of 1.2 V and −0.2 V vs RHE, respectively. **e** Radial intensity profiles as a function of applied electrode potential. **f** Plot of average d-space values determined from the Pd/PdH$_x$(111) diffraction peak fitting as a function of electrode potential, with the dashed blue and red lines representing the theoretical values for the metallic Pd and β-phase PdH$_x$, respectively, obtained from crystallographic information file (cif) databases. The d-space values were calculated based on the center of the electron diffraction peak using the Crystal Box software with an accuracy of ±0.02 Å.

to detachment. A similar observation was previously shown for carbon-supported Pt and Pd catalysts investigated using identical location TEM[59,60]. Figure 4e provides a schematic depiction of the Pd/PdH$_x$ transformations and particle degradation processes that were identified using the unique insights provided by LP-(S)TEM imaging under CO$_2$R conditions. To confirm the changes observed in the Pd/PdH$_x$ particles were not influenced by contamination of the working electrode from the Pt-based counter electrode[61], EDX mapping of the working electrode was conducted (Supplementary Fig. 14) and indicated the particles consisted of only pure Pd.

### Electrochemical CO$_2$R Activity and Selectivity

The electrochemical CO$_2$R performance of Pd particles prepared by the same technique (electrodeposition) and with similar structural properties to those characterized by in-situ LP-(S)TEM/SAD was evaluated using large-format electrodes which enabled quantification of CO$_2$R activity and selectivity metrics. Figure 5a shows the Faradaic

efficiency (selectivity) and current density (activity) of the Pd/PdH$_x$ particles towards electrochemical CO$_2$R tested by 1-hour chronoamperometry measurements at electrode potentials between -0.1 to -0.5 V vs. RHE. The Pd/PdH$_x$ particles showed the highest selectivity towards formate at -0.2 V vs RHE, with a Faradaic efficiency of 94%. At more negative applied potentials (-0.3 to -0.5 V vs. RHE), the selectivity of the Pd/PdH$_x$ particles towards formate was reduced significantly, showing a Faradaic efficiency of only 6% at -0.5 V vs. RHE. The major products formed at this potential were H$_2$ (FE of 60 %) and CO (FE of 30%). Tafel plots of the partial current densities for H$_2$, CO, and formate versus potential are shown in Supplementary Fig. 15. An increase in partial current density towards formate is observed from -0.1 to -0.2 V vs RHE, as would be expected for a reaction following Tafel's behavior. However, at potentials more negative than -0.2 V vs RHE, the partial current density towards formate plateaus and then decreases substantially, indicating catalyst surface poisoning or a shift in the electrochemical CO$_2$R mechanisms that result in modulated product

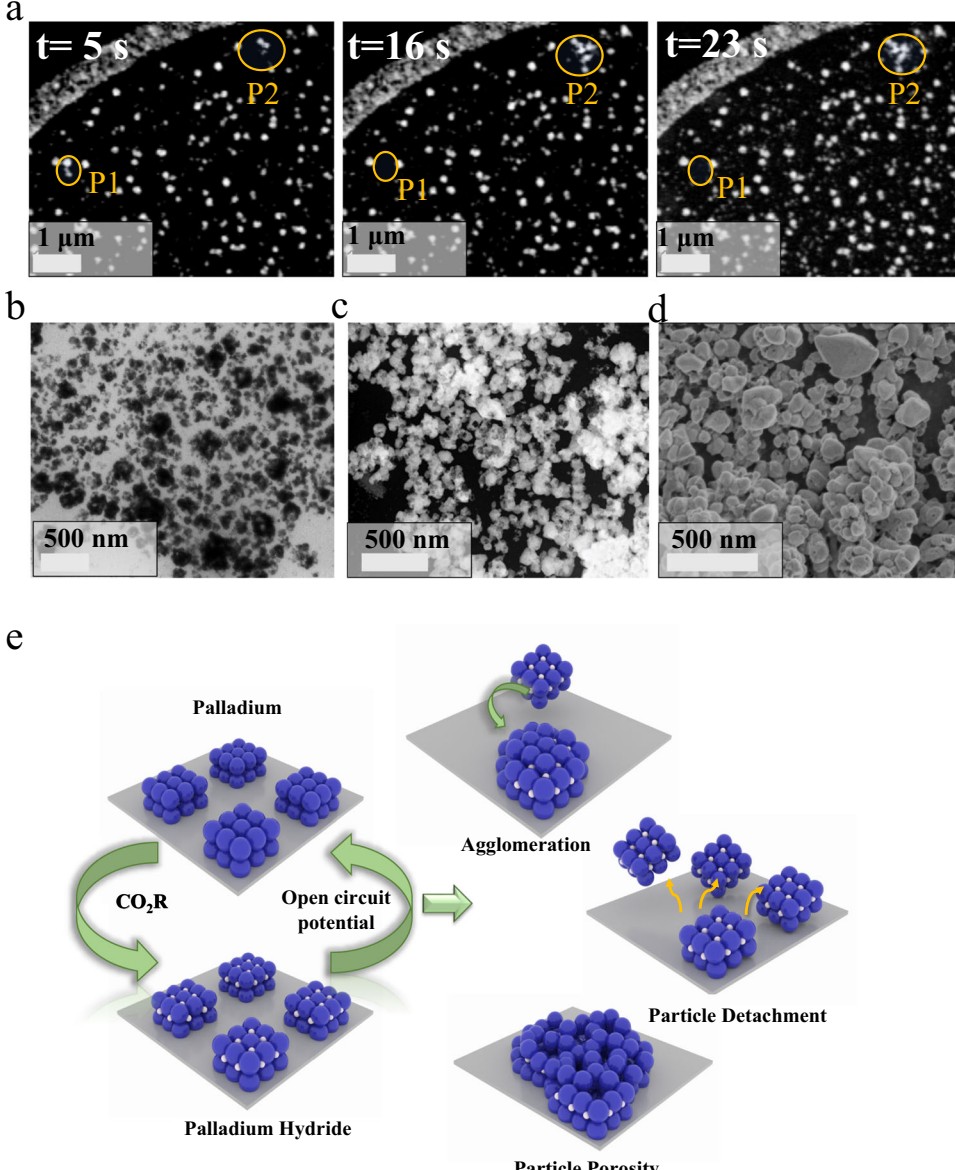

**Fig. 4 | Morphological changes occurring in Pd/PdH$_x$ particles under electrochemical CO$_2$ conditions. a** In-situ LP-HAADF-STEM snapshots extracted from Supplementary Movie 3 illustrating Pd/PdH$_x$ particle migration, agglomeration, and detachment from the glassy carbon working electrode at different electrode potential hold times under an applied potential of −0.2 V vs. RHE (yellow circles indicate some areas where the agglomeration and detachment are more obvious). **b** TEM **c** HAADF-STEM, and **d** SEM images of the Pd particle morphology on the in-situ glassy carbon electrode after in-situ imaging under CO$_2$ electrolysis conditions. **e** Schematic depiction of the morphological evolution of Pd/PdH$_x$ catalysts revealed by in-situ LP-TEM measurements.

selectivity. This dramatic shift in CO$_2$R selectivity coincides with the increased intercalation of protons into the PdH$_x$ structure, with more insight into these phenomena analyzed by DFT and discussed in more detail in the proceeding sections.

## Pd Surface Recovery After CO$_2$R

The presence of *CO and *H species (* indicates adsorbed species) on the surface of Pd/PdH$_x$ under CO$_2$R conditions has been shown to influence the activity, selectivity, and structural evolution of the catalyst[9,12,15]. To investigate the presence of these species, electrode potential holds under CO$_2$R conditions were carried out on the Pd-decorated large-format electrode followed by cyclic voltammetry measurements to determine the subsequent electrochemical response. Initially, the electrodes were held for varying amounts of time at different electrochemical CO$_2$R-relevant potentials in CO$_2$ saturated 0.1 M KHCO$_3$. Without relaxing to open circuit potential, the

electrode potential was then swept by linear sweep voltammetry up to 1.2 V vs RHE, which enabled us to observe *CO stripping peaks to provide insight into *CO surface poisoning phenomena (Fig. 5b). Following this sweep, cyclic voltammetry was conducted until a steady state profile was collected, with the steady state profile denoted in Fig. 5b as the "baseline CV" measurement. For the linear sweep voltammetry measurements immediately following the chronoamperometric hold under CO$_2$R conditions (3 min hold at potentials from -0.1 to -0.5 V vs RHE), two oxidation peaks were observed and likely attributed to the oxidation of adsorbed surface *CO species or the desorption/deintercalation of H species[62–67]. For example, a 3-minute electrode potential hold at -0.1 V vs RHE led to a subsequent linear sweep voltammetry measurement with a prominent oxidation feature at electrode potentials <0.5 V vs RHE, attributed to the desorption/deintercalation of H species. A buildup of adsorbed *CO species was also indicated by the subtle oxidation peak observed at ca. 0.9 V vs

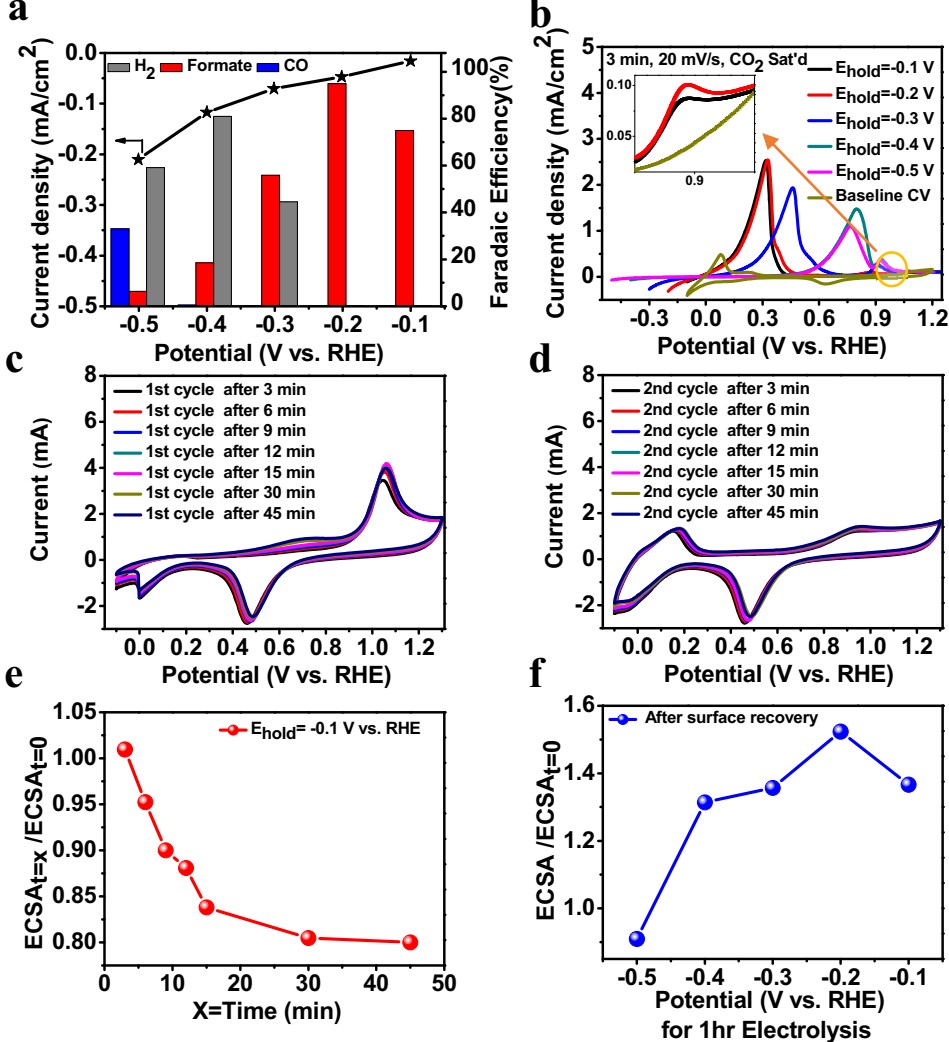

**Fig. 5 | Electrochemical CO₂R selectivity and surface recovery of Pd particles after poisoning by adsorbed \*CO species. a** Faradaic efficiencies (left y-axis) and partial current densities (right y-axis) for the production of formate, H₂ and CO. **b** Positive linear sweep voltammetry following a 3 min electrode potential hold at various CO₂R electrode potentials, along with baseline cyclic voltammetry curves collected in CO₂ saturated 0.1 M KHCO₃ electrolyte with a scan rate of 20 mV/sec. **c** and **d** Cyclic voltammetry measurements including the 1ˢᵗ and 2ⁿᵈ cycle following varying durations of an electrode potential hold at −0.1 V vs. RHE at scan rate 50 mV/sec. **e** ECSA estimations using electrochemical double layer capacitance measurements immediately following an electrode potential hold at −0.1 V vs RHE for varying time durations (x) normalized by the original ECSA of the catalyst particles (i.e., ECSA$_{t=x}$/ ECSA$_{t=0}$), demonstrating the effect of \*CO poisoning on the ECSA. **f** ECSA estimations following a 1 hr electrode potential hold at various electrode potentials and subsequent cyclic voltammetry sweeps to strip \*CO species and recover a clean Pd surface. ECSA values are normalized by the original ECSA of the catalyst particles (i.e., ECSA/ECSA$_{t=0}$).

RHE (shown at higher magnification in the inset of Fig. 5b). Applying more negative electrode potentials during the chronoamperometry potential hold, the subsequent linear sweep voltammetry measurements showed that the H desorption/deintercalation peaks shifted to higher potentials, likely arising from the higher concentration of accumulated adsorbed \*CO species at more negative electrode potentials as claimed previously[67], as well as an increased amount of H absorbed into the PdH$_x$ lattice as demonstrated by in-situ LP-TEM/SAD measurements.

To investigate further, the same measurements were carried out in N₂ saturated 0.1 M KHCO₃ to gain insights into the electrochemical response of the Pd/PdH$_x$ particles in the absence of significant concentrations of CO₂ (and subsequent buildup of adsorbed \*CO species), with results shown in Supplementary Fig. 16. Please note, the conversion between the Ag/AgCl reference electrode used to carry out these measurements and the RHE scale took into account the pH difference between these two experimental conditions (pH of 6.8 for CO₂ purged electrolyte versus 8.3 for N₂ purged). After a 3-min electrode potential

hold at potentials ranging from -0.1 to -0.5 V vs RHE in the N₂ saturated electrolyte, only one oxidation peak at ≤ 0.5 V vs RHE was observed in the subsequent linear sweep voltammetry measurement, attributed to desorption/deintercalation of H from PdH$_x$. Substantial shifts in the electrode potential of these oxidation features were not observed when more negative chronoamperometry potentials were applied, providing evidence that the shifts in the H-desorption/deintercalation peaks observed in the case of CO₂ saturated 0.1 M KHCO₃ were largely due to the presence of the adsorbed \*CO species and to a lesser extent from the increased concentration of absorbed H in the PdH$_x$ structure.

For electrochemical CO₂R measurements, an increased current density was observed at more negative electrode potentials (Fig. 5a). Over the course of the electrode potential holds used to measure CO₂R activity and selectivity, a decrease in the current density for CO₂R was observed with time (Supplementary Fig. 17), potentially due to gradual poisoning of the Pd/PdH$_x$ surface with \*CO. In addition to surface poisoning by \*CO, detachment and agglomeration of Pd/PdH$_x$ particles shown by in-situ (S)TEM measurements (Fig. 4) represents

another mechanism by which the active surface available for the electrochemical $CO_2R$ can become diminished. To gain insight into these simultaneous processes, electrochemically active surface area (ECSA) values were estimated using double-layer capacitance measurements at various stages throughout the course of the chronoamperometry hold and subsequent linear sweep voltammetry measurements detailed in the previous paragraph and shown in Fig. 5b. ECSA values were estimated by conducting cyclic voltammetry measurements between 0.2 and 0.4 V vs RHE at varying scan rates as outlined in more detail in Supplementary Note 5 and Fig. 18 of the supplementary information. This route was selected for ECSA estimation as hydrogen underpotential deposition ($H_{upd}$) measurement could not provide reliable measurements as a significant portion of the current measured in the potential region attributed to $H_{upd}$ for Pd was due to either H adsorption/intercalation or desorption/deintercalation. Moreover, it is important to note that reliable ECSA measurements in the in-situ LP-TEM electrochemical cell were difficult to conduct owing to the small size of the electrodes, which led to very small currents obtained during CV measurements that were significantly impacted by the double-layer capacitance of the underlying glassy carbon electrode. The ECSA for the electrodeposited $Pd/PdH_x$ particles was estimated before ($ECSA_{t=0}$) and after chronoamperometric potential holds at -0.1 V vs RHE in $CO_2$-saturated 0.1 M $KHCO_3$ for durations ranging from 3 to 45 mins ($ECSA_{t=3 \text{ to } t=45}$). Following ECSA measurements, cyclic voltammetry scans from -0.1 to 1.3 V vs RHE were applied to remove adsorbed *CO species and restore the "clean" Pd surface. Results of this measurement are shown in Fig. 5c, demonstrating a *CO stripping peak between ca. 0.9 and 1.1 V vs RHE with an increased magnitude of the peak observed with increasing electrode potential hold times. The H desorption/deintercalation are not observed in these cyclic voltammetry measurements as adsorbed/intercalated H species were removed at the electrode potentials applied during the measurements used for ECSA estimation. After the first cycle where *CO species removal was observed (Fig. 5c), subsequent cyclic voltammetry cycles showed negligible differences to each other indicating that the electrode had reached steady state and a pristine Pd surface was recovered. Figure 5d therefore plots the 2nd cycle as a representative example.

To track the impact of *CO poisoning on the ECSA of the $Pd/PdH_x$ particles during $CO_2R$, the $ECSA_{t=x}/ECSA_{t=0}$ was estimated (Fig. 5e), where time (t) indicates the duration of the electrode potential hold at -0.1 V vs RHE. When the electrode potential hold period was prolonged from 3 min to 45 mins, the $ECSA_{t=x}/ECSA_{t=0}$ ratio was reduced from 1.01 to 0.80, demonstrating an approximately 20% loss in surface area. This reduction in ECSA could be recovered using cyclic voltammetry to strip *CO and restore the pristine Pd surface, indicating the loss in ECSA observed immediately following the electrode potential hold likely arose due to *CO poisoning. It was then desirable to identify ECSA changes following longer electrode potential holds under $CO_2R$ conditions. 1 hr electrode potential holds were therefore conducted sequentially at increasingly more negative electrode potentials, starting at -0.1 V vs RHE and proceeding in increments of 100 mV down to -0.5 V vs RHE. Between each 1 hr electrolysis hold, repeated cyclic voltammetry measurements were conducted to clean the Pd surface and ECSA values were measured by double-layer capacitance to calculate the $ECSA/ECSA_{t=0}$ ratios shown in Fig. 5f. The electrolyte was also replaced with fresh electrolyte to remove possible contaminants or liquid phase $CO_2R$ products that could impact subsequent measurements before subsequent electrode potential holds and electrochemical measurements were applied. The calculated $ECSA/ECSA_{t=0}$ after a 1 hr electrode potential hold at -0.2 V vs RHE and cyclic voltammetry cleaning showed the highest value of 1.5. This increase was attributed to the introduction of porosity into the $Pd/PdH_x$ particles that occurred over all regions of the electrode as revealed by ex-situ HAADF-STEM imaging of the electrodes after $CO_2R$ (Supplementary

Fig. 12) as discussed previously. At more negative electrode potential holds from -0.3 to -0.5 V vs. RHE, the calculated $ECSA/ECSA_{t=0}$ decreased from 1.3 to 0.9, respectively. It is interesting that the normalized ECSA decreases (after surface recovery) at more negative potential, despite the observation of redeposition of smaller Pd particles under cathodic potentials (Fig. 4a). This subsequent net decrease in ECSA (observed after *CO removal) is likely due to the detachment of the $Pd/PdH_x$ particles from the electrode surface and some particle agglomeration/growth observed via in situ LP-TEM as discussed previously. Similar particle detachment morphological changes were also observed on the large-format electrodes after a one hour electrode potential hold at -0.5 V vs RHE (Supplementary Fig. 19 and Supplementary Fig. 20), reinforcing the fact that $Pd/PdH_x$ particle detachment was prevalent at these conditions and responsible for the observed ECSA decrease.

## Mechanistic Insight into $CO_2R$ Activity and Selectivity Through Density Functional Theory

DFT calculations were performed to provide an understanding of the structure-property-performance trends observed via in-situ LP-(S)TEM correlated with $CO_2R$ activity/selectivity measurements on electrodeposited $Pd/PdH_x$ particles. The constant potential methodology implemented in the Solvated Jellium Method (SJM)[68] was applied, enabling the simulation of adsorbed $*CO_2$, which is only possible when including explicit charging of the electrode. The (111) and (100) facets of the fully hydrogenated β-phase $PdH_x$ were studied using input from the in-situ LP-TEM and SAD studies that identified the presence of this phase at potentials below -0.2 V vs RHE. Hydrogen atoms were located in the octahedral sites of bulk Pd, corresponding to hollow sites on the surface of Pd. The lattice parameters of the β-$PdH_x$ bulk structure were optimized, leading to a 0.12 Å increase in the d-spacing compared to metallic Pd, in line with the experimental observations presented in Fig. 3f. Supplementary Fig. 21 shows the calculated adsorption-free energies of $H^+$ from solution on a $PdH_x$(111) surface at varying *H coverages. The calculated binding energies suggest an incomplete monolayer of *H is present at 0 V vs. RHE, with nearly thermoneutral binding energies up to a coverage of ¾. At -0.3 V vs. RHE, a complete monolayer is present.

Figure 6a shows the calculated free energy pathways for electrochemical $CO_2R$ towards formate and CO occurring on the $PdH_x$(111) facet, while results for the $PdH_x$(100) facet are shown in Supplementary Fig. 22. Particularly, the $PdH_x$(100) facet is found to be poisoned with *CO adsorbates under reaction conditions and hence not active in producing CO or formate. A similar poisoning effect could be expected for less-coordinated surface terminations, making the most thermodynamically stable (111)-facet the most likely surface facet to contribute to the high activity towards formate production observed at low overpotentials. All subsequent calculations discussed were, therefore, performed on the $PdH_x$ (111) facet. The mechanism of $CO_2R$ to produce CO was found to pass through a (bent) $*CO_2$ intermediate bound to $PdH_x$ on a Pd top-site via the carbon atom, as shown in the atomic scale schematic at the top of Fig. 6a. On the other hand, the mechanism of $CO_2R$ toward formate does not proceed through a stable $*CO_2$ intermediate. Rather, $CO_2$ is found to react with surface-bound hydrogen in a Heyrovsky-like mechanism[69] after being activated close to the $PdH_x$ surface (Supplementary Fig. 23a). This mechanism towards producing formate on $PdH_x$ (and possibly other metal hydrides) is in stark contrast to proposed mechanisms of formate production on oxophilic post-transition metals (such as Pb[70] and Sn[71]) that have been proposed as capable of stabilizing $CO_2$ binding via the oxygen atoms (*OCO) followed by protonation to produce formate[71-73]. Conversely, stabilizing the *OCO intermediate on $PdH_x$ in the simulations was not successful, even upon a rigorous sampling of various binding motifs. The fact that the key intermediate in formate production, *OCO, does not bind strongly on $PdH_x$ suggests that the hydrogenation of the central carbon atom occurs via the weakly bound

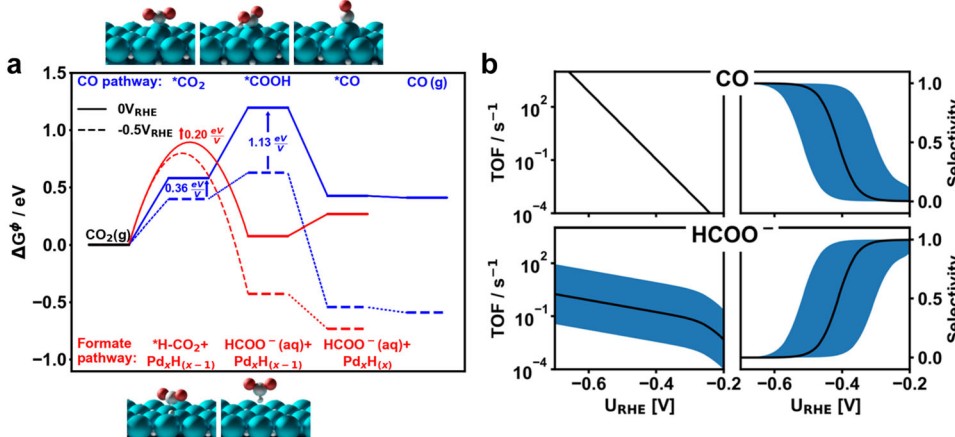

**Fig. 6 | Density functional theory and micro-kinetic modeling analyses.**
**a** Calculated DFT (constant potential) free energy diagram of the reaction pathways for $CO_2R$ towards formate and CO, including the calculated reaction barrier of $CO_2$ hydrogenation (*H-$CO_2$) plotted at 0 and $-0.5$ V vs. RHE (solid lines and dashed lines, respectively). The optimized geometries of the associated reaction steps are shown explicitly next to the labels. **b** Turnover frequency and selectivity results obtained from micro-kinetic modeling using the results from DFT analyses as inputs. The shaded blue region indicates the uncertainty in TOF and selectivity calculations that arise if the barrier for $CO_2$ surface hydrogenation varies by $+/- 0.1$ eV.

(sub-)surface hydrogen atoms. The reaction mechanism involving $CO_2$ hydrogenation towards producing formate was identified as an outcome of the transition state searches starting from both solvated (planar) and adsorbed (bent) $CO_2$. Both starting states for the reaction (i.e., solvated or adsorbed $CO_2$) led to the same transition state, where partially activated $CO_2$ reacts with a (sub)surface *H (*H-$CO_2$). The potential response at the transition state was calculated to be 0.2 eV/V, while the majority of the charge injection leading to the formate occurred after the transition state (Supplementary Fig. 23b).

The formation of *COOH was identified as the bottleneck towards producing CO at relevant potentials with the formation calculated to exhibit a potential response of 1.13 eV/V (Fig. 6a). As the energetics for forming *COOH are more strongly dependent on potential than the formation of *H-$CO_2$, CO production becomes energetically favored at increasingly negative electrode potentials. At 0 V vs. RHE, however, *H-$CO_2$ is 0.30 eV more stable than *COOH, rendering formate the preferentially formed product at this electrode potential. It should be noted that in our model the reaction rate of *$CO_2$ to *COOH is only limited by the thermodynamic barrier, as the negative partial charge on the O-end of *$CO_2$ enables facile oxygen protonation[74]. However, the appearance of a kinetic barrier for the step from *$CO_2$ to *COOH would not change the qualitative behavior of a larger stabilization of the CO-path with more negative potentials. The 0.36 eV/V potential response of *$CO_2$ is still larger than the 0.2 eV/V of *H-$CO_2$, and the barrier from *$CO_2$ to *COOH cannot exhibit a lower potential response than either of the end states.

Based on the described reaction energetics, a microkinetic model was constructed (Fig. 6b). The calculated turnover frequencies (TOF) towards formate outweigh the TOF towards CO at electrode potentials between -0.2 and -0.35 V vs RHE. At more negative potentials, the TOF towards both CO and HCOO⁻ increases, although the increase in TOF for CO is much more drastic. The selectivity for CO (TOF towards CO divided by the sum of the TOFs towards both CO and HCOO⁻) increases as a result of the strong potential response calculated for *COOH as described above. Therefore, this analysis indicates the $CO_2R$ selectivity towards CO should increase at more negative potentials owing to the electrode potential-dependent energetics of the reaction-relevant species. The results of this microkinetic modeling are in agreement with experimental $CO_2R$ measurements that indicate a shift in selectivity from formate towards CO at increasingly negative potentials (Fig. 5a). A quantitative agreement between experiment and theory is generally not expected[75], thus calculations were performed to understand the sensitivity the calculated microkinetic models have on the

calculated free energy values of *H-CO. Figure 6b shows blue shaded areas that represent the variation in TOF and selectivity that would be expected with a difference in the calculated *H-CO free energy of ±0.1 eV. This range in the energetic uncertainty leads to a ca. 0.2 V difference in the electrode potential at which a $CO_2R$ selectivity change from producing HCOO- to producing CO would be expected. However, as discussed above, this uncertainty does not alter the qualitative finding that $PdH_x$ produces formate almost exclusively at low overpotentials, followed by a sharp change in selectivity towards CO production at more negative potentials.

## Discussion

The morphological and phase structure conversions occurring in Pd/$PdH_x$ catalyst under electrochemical $CO_2R$ conditions were revealed by in-situ LP-(S)TEM characterization and supplemented by ex-situ post-$CO_2R$ characterization of the electrodes. As shown in Fig. 4e, three primary changes to the Pd/$PdH_x$ were observed under $CO_2R$ conditions: (1) Particle agglomeration; (2) Particle detachment from the electrode surface; and (3) Hollowing out of the particles to form a sponge-like porous morphology.

To an extent, the particle agglomeration observed could follow Ostwald ripening or other sintering mechanisms. Localized in-situ particle tracking measurements could be applied to understand this process of particle sintering as well as the underlying kinetics[9], which is outside the scope of the present work. The particle agglomeration observed in the in-situ LP-(S)TEM measurements was found to occur by particle detachment from the electrode surface and subsequent deposition on another region of the electrode (Fig. 4a) that was most prevalent at more negative electrochemical $CO_2R$ conditions. It is speculated that particle detachment was largely induced by mechanical forces that arose due to the absorption of increased amounts of H into the $PdH_x$ lattice at increasingly negative electrode potentials, which led to significant volume expansion (and contraction upon conversion back to metallic Pd). This could lead to mechanical instabilities at the catalyst/electrode interface, causing the detachment of catalyst particles[58]. It should also be noted that mechanical agitation from the formation of bubbles ($H_2$ and/or CO) at increasingly negative potentials could also influence particle detachment; however, there was no direct observation of this by in-situ LP-(S)TEM measurements. The hollowing of Pd/$PdH_x$ to form porous sponge-like particles also occurred under $CO_2R$ conditions, likely due to thermodynamic driving forces[76]. Additionally, the evolution of porosity through morphological changes could arise due to adsorbate (i.e., *CO) induced

restructuring[13,77]. The presence of *CO species on the surface of Pd/PdH$_x$ under CO$_2$R conditions was demonstrated in this work (Fig. 5b), with DFT calculations, suggesting *CO poisoning to be a facet dependent occurrence on Pd/PdH$_x$.

The phase structure of Pd/PdH$_x$ was observed as a function of electrode potential under CO$_2$R conditions by in-situ LP-TEM/SAD. At potentials between -0.1 and -0.2 V vs RHE, the particles were in a mixed α/β-phase PdH$_x$, with the complete formation of the β-PdH$_x$ phase observed at more negative potentials. Interestingly, this conversion of mixed-phase α/β-PdH$_x$ to β-phase PdH$_x$ coincided with a CO$_2$R selectivity shift from formate (at lower overpotentials) towards CO (at higher overpotentials), suggesting that the phase transformation may be underlying the catalytic trends. However, DFT calculations guided by the results of in-situ LP-TEM and SAD measurements and coupled with micro-kinetic modeling indicated the primary reason for the CO$_2$R selectivity change was due to the electrode potential dependent thermodynamic energetics of adsorbed reactive intermediate *COOH in the case of CO or *H-CO$_2$ in the case of formate. For the production of formate on the phase structures observed under reaction conditions, the hydrogenation of the C atom in CO$_2$ by (sub)surface *H was identified as a key step, in contrast to the formate production mechanism suggested previously for oxophilic metals such as Sn and Bi, whereby the CO$_2$ molecules were found to adsorb on the catalyst surface via the O atoms[71–73]. Formate production through the reaction of CO$_2$ with *H has been postulated in the past based on electro-kinetic measurements in a three-electrode (two compartment) electrochemical cell in which the concentrations of reaction relevant species were systematically verified[9]. However, the phase structure(s) and morphology of the Pd/PdH$_x$ catalysts that are present under reaction conditions were not identified through in situ characterization, and furthermore the reaction energetics and identity of intermediate reaction species was not rigorously explored through the combination of in-situ catalyst characterization measurements correlated with DFT and micro-kinetic modeling. The direct participation of (sub)surface *H in CO$_2$R towards formate suggests that PdH$_x$ catalysts with an increased *H availability as a reactant (and a reduction in the binding strength of *H) is a desirable catalyst design principle. However, at -0.2 V vs RHE where the highest selectivity towards formate is observed, the catalyst is in a mixed α/β-PdH$_x$ phase. At lower electrode potentials complete conversion to the β-PdH$_x$ phase and increased saturation of the lattice with hydrogen does not coincide with an increase in activity/selectivity towards formate. Instead, the opposite trend is observed and DFT alongside micro-kinetic calculations shows that at more negative electrode potentials, the formation of the *COOH intermediate (for producing *CO) becomes energetically more favorable than the formation of the *H-CO$_2$ intermediate for producing formate. Therefore, the selectivity change of the CO$_2$R from producing formate toward CO at more negative electrode potentials is attributed to thermodynamic changes to the energetics of the reaction and not due to the phase transformation of increased content of absorbed H in the PdH$_x$ structures present under reaction conditions. It is also important to note that the heterogeneity of surface structures arising from the morphology of electrocatalysis particles should not be overlooked. Herein we have primarily studied the most thermodynamically stable (111) surface as it is the weakest-binding planar surface termination and therefore least likely to be poisoned, while also having the highest propensity for Pd-H bond breaking that is central for the production of formate. Thus we postulate that the (111) surface is representative of the structures and electrocatalysis trends observed in this work. Detailed screening of various surface terminations including under-coordinated surfaces consisting steps and defects, or potentially more interestingly, over-coordinated surfaces that weaken interactions with potential poisoning species and reaction intermediates is outside the scope of this study. The scientific community is encouraged to consider and apply the design principles outlined herein in future efforts to design and understand Pd/PdH$_x$ catalysts, especially

considering that the morphology of the Pd/PdH$_x$ particles and microstructure of the electrode will evolve under electrochemical CO$_2$R conditions as observed by the in situ LP-TEM measurements.

The mechanistic insights produced in this work enables us to propose design principles for alternative catalysts with PdH's extraordinary ability to selectively produce formate. First, for the identified mechanism, the presence of (sub-)surface hydrogen at reaction conditions is essential, suggesting catalysts with negative formation energies of *H. However, the binding of hydrogen should be weak as the formation of *H-CO$_2$ correlates negatively with it. Thus, an ideal binding energy would range close to a net zero free energy change for hydrogen adsorption. Simultaneously, destabilizing *COOH increases the potential window where formate can be produced selectively over CO. It has previously been found that the formation of *H and *COOH are correlated energetically (commonly referred to as a scaling relation)[78]. This poses a fundamental limitation to formate selectivity as the two descriptors cannot be altered independently and the tendency to increase the availability of *H, would simultaneously stabilize *COOH, increasing the activity of CO production. Breaking this scaling might be key to developing efficient and selective electrocatalysts to produce formate within CO$_2$ electroreduction following the same mechanism as Pd.

In conclusion, in-situ LP-(S)TEM and SAD measurements were conducted on electro-deposited Pd/PdH$_x$ catalysts to identify morphological and phase structure changes occurring in these materials under electrochemical CO$_2$R conditions. Under electrochemical CO$_2$R conditions, the Pd/PdH$_x$ catalysts underwent morphological changes, including (i) particle agglomeration; and (ii) formation of a porous sponge-like morphology likely arising from adsorbate (i.e., *CO) induced restructuring. Additionally, particle detachment from the electrode surface was observed, likely due to mechanical agitation induced by the process of interconversion between the metallic Pd and PdH$_x$ phase(s). Electrochemical CO$_2$R activity and selectivity measurements revealed that formate was produced almost exclusively at -0.2 V vs RHE, whereby the production of H$_2$ and CO became prominent at more negative potentials. Correlation with LP-TEM-SAD measurements showed this selectivity shift coincided with increased H absorption into the PdH$_x$, forming a β-PdH$_x$ phase. By coupling in-situ structural analysis and electrochemical evaluation of the Pd/PdH$_x$ catalysts with DFT calculations and micro-kinetic modeling, it was demonstrated the CO$_2$R selectivity from formate to CO/H$_2$ changes occurred due to potential-dependent reaction energetic changes and not due to the observed PdH$_x$ lattice expansion. DFT calculations revealed the reaction mechanism towards formate on β-PdH$_x$ involved hydrogenation of the C atom in the CO$_2$ molecule by (sub)surface *H present in PdH$_x$. This contrasts the formate production mechanism suggested previously for oxophilic metals where the CO$_2$ molecule is likely to adsorb on the catalyst surface via its oxygen atoms and is subsequently protonated. At more negative electrode potentials, the *COOH intermediate for producing CO was stabilized in comparison to the *H-CO$_2$ intermediate for producing formate, explaining the dramatic shift in selectivity from nearly-exclusive production of formate at -0.2 V vs RHE to the production of CO/H$_2$ at -0.5 V vs RHE. This work, therefore, provides mechanistic insight into the electro-catalytic mechanisms of CO$_2$R occurring on Pd-based catalysts that can be applied to understand and guide future catalyst designs. Furthermore, in-situ LP-(S)TEM including SAD has been demonstrated as a powerful technique for gaining unprecedented insight into the morphological and phase structure changes occurring during the PdH$_x$/Pd interconversion process and specifically applied for catalytically relevant materials under electrochemical CO$_2$R conditions.

## Methods

### Pd particle electrodeposition using in-situ LP-TEM electrochemical reactor

The electrochemical setup for Pd electrodeposition onto the in-situ electrochemical TEM sample holder is shown in Supplementary Fig. 1.

The Poseidon Select (Protochips) in-situ TEM sample holder was utilized, whereby the micro-chip electrochemical cell mounted in the tip of the TEM holder consists of a Pt reference and counter electrode, and a glassy carbon working electrode (Supplementary Fig. 1). The microchip reactor consists of a top and bottom chip that are sealed together with a gasket and fastened with screws. Both the bottom and top chips contain a thin silicon nitride ($SiN_x$) membrane viewing window that enables electron transmission for in-situ TEM measurements. As received, both the top and bottom chips are coated with a protective photoresist layer to prevent $SiN_x$ membrane damage. The protective photoresist layer was removed prior to the two-step rinse process, whereby chips were submerged first in acetone and then methanol, each for 2 min. To enhance the hydrophilicity of the chips, a plasma cleaning (Gatan plasma system model 950 advanced plasma, with $Ne/H_2/Ar$ gas mixture and operating at 15 W) was used. The process was performed for 2 min for the small E-chip and 30 s or less for the large E-chip as the excessive plasma cleaning could damage the glassy carbon electrode. Following assembly of the micro-chip electrochemical cell, a liquid solution of 5 mM $H_2PdCl_4$ with 0.015 M HCl was introduced at a flow rate of 5 μL/min through the microfluidic channels of the sample Poseidon Select holder using an external syringe pump. Once the solution was introduced, electrochemical chronoamperometry was carried out at 0.2 V vs. RHE using a floating potentiostat (Gamry Reference 600+) for 120 s to ensure the electrodeposition of a sufficient amount of Pd particles on the working electrode. Following electrodeposition, the in-situ TEM holder was purged with Millipore water to remove the electrodeposition solution.

### In-situ (S)TEM measurements under electrochemical $CO_2R$ conditions

In-situ electrochemical (S)TEM liquid cell measurements were conducted to investigate the phase and structural transformations of electrodeposited Pd particle catalysts under $CO_2R$ conditions. For all in-situ TEM experiments, to avoid $SiN_x$ window bulging due to the pressure difference between the electrochemical micro-chip cell and the vacuum in the TEM column, a perpendicular (crossed configuration) window strategy was utilized as recommended by previous studies[79]. After electrodeposition of Pd and purging of the Pd salt solution by Millipore water, the Millipore water was replaced by flushing the electrochemical cell TEM holder with freshly prepared $CO_2$ saturated 0.1 M $KHCO_3$ solution at a flow rate of 5 μL/min. Confirmation that the 0.1 M $KHCO_3$ had entered the sample holder was indicated when the open circuit potential was stabilized. Leak checking of the in-situ TEM sample holder was performed before insertion into the microscope using a custom-designed vacuum pump station. To establish a baseline, in-situ LP-TEM imaging and select area electron diffraction (in-situ LP-TEM/SAD) measurements were performed at different times: 1, 5,7, and 10 min before applying any electrode potential. After these measurements, chronoamperometry at different applied potentials in the range of 1.3 to -0.2 V vs. RHE was applied for 60 s at each potential, during which time in-situ LP TEM imaging and in-situ LP-TEM/SAD patterns were collected. Between each chronoamperometry experiment, a potential of 1.2 V vs. RHE was applied for 60 s to recondition the particles to be in metallic Pd form, thereby avoiding any issues pertaining to $Pd/PdH_x$ transformation hysteresis[80]. Detailed information about in-situ LP-TEM/SAD analysis and beam dose calculations are included in Supplementary Note 1.

### Pd electrodeposition on large-format glassy carbon electrodes for $CO_2R$

Electrodes to test the electrochemical $CO_2R$ activity and selectivity of electro-deposited Pd were prepared using a large-format glassy carbon electrode with dimensions of 2 cm by 5 cm. Electrodeposition of Pd particles was performed by chronoamperometry at 0.2 V vs. RHE for 120 s in 5 mM $H_2PdCl_4$ mixed with 0.015 M HCl. A Pt foil counter electrode and Ag/AgCl reference electrode that was calibrated and converted to the RHE scale were used. After electrodeposition, the electrode was rinsed carefully with Millipore water and dried at room temperature under $N_2$ gas flow.

### Electrochemical $CO_2R$ activity/selectivity measurements

Electrochemical $CO_2R$ activity and selectivity of the electrodeposited Pd catalyst were investigated using a custom-built electrochemical cell (Supplementary Fig. 2) reported on previously[81], which was designed and improved upon to provide high sensitivity for $CO_2R$ product detection and quantification. On-line gas chromatography (SRI Multigas #5) was used to detect/quantify gas products while liquid products were quantified using the Bruker AVIII 700 NMR available at McMaster University's Nuclear Magnetic Resonance Facility. A mass flow control unit (pMFC, MKS Instrument) was used to maintain a $CO_2$ flow rate of 20 sccm through the catholyte chamber throughout the entire course of the reaction. A Pt foil was used as the counter electrode and Ag/AgCl as the reference electrode, which was calibrated and converted to the RHE scale by measuring the open circuit potential of the Ag/AgCl versus an in-house designed RHE. $CO_2R$ electrolysis measurements were conducted by chronoamperometry at a potential between -0.1 V to -0.5 V vs. RHE for one hour each, while cyclic voltammetry measurements were conducted at 50 mV/s before and after chronoamperometry. The geometric surface area of the large format electrode exposed to the electrolyte was 5.6 $cm^2$.

### Materials

Potassium bicarbonate (ACS reagent, 99.7%), palladium (II) chloride (99.9%), and hydrochloric acid (ACS reagent, 37%), were purchased from Sigma Aldrich and used without any further purification.

### Materials characterization

To investigate the morphology and composition of Pd electrocatalysts immediately after electrodeposition and after $CO_2R$ testing, optical microscopy (CLEMEX, Axioplan 2 imaging), scanning electron microscopy (JEOL JSM-7000F SEM), high-resolution transmission electron microscopy (HRTEM), high-angle annular dark-field scanning transmission electron microscopy (HAADF-STEM) imaging, along with energy dispersive X-ray (EDX) mapping were carried out. All TEM and HAADF-STEM imaging were performed using an image-corrected FEI Titan 80-300LB operating at 300 kV and a Thermo Scientific Talos 200× operating at 200 kV available at the Canada Center for Electron Microscopy (CCEM) at McMaster University.

### Computational details

The reported DFT-based (constant potential) energies were calculated using the constant-potential mode of SJM[68] implemented in GPAW[82,83]. A real-space grid basis set was applied with a grid spacing of 0.18 Å. The BEEF-vdW functional[84] was applied for approximating the XC contributions. All slab calculations were conducted with periodic boundary conditions parallel to the slab surface and a dipole correction in the direction perpendicular to the surface was applied. $3 \times 4 \times 4$ supercells were used, with the bottom two layers being constrained to the bulk lattice constants of Pt and PtH, respectively. Monkhorst−Pack k-point grids of $4 \times 4 \times 1$ and $4 \times 3 \times 1$ were applied for palladium and palladium hydride structures, respectively. The setup used a Fermi smearing of 0.1 eV/$k_B$. Forces were converged to 0.03 eV/Å and 0.05 eV/Å for stable intermediates and transition states, respectively.

SJM uses an effective potential cavity solvation model implemented into GPAW by Held and Walter[85]. The parameters used were: Bondi's van der Waals radii[86], the strength of the repulsion at the atomic radii controlling the cavity size, u0 = 0.18 eV, surface tension 0.001148 Pa*m (both taken from[85], (maximal) dielectric constant (ϵ) = 78.36 and temperature = 298.15 K. The tolerance for the electrode potential deviation from the target potential was set to 10 mV.

All possible adsorbate binding configurations were sampled using the CatKit Surface module[87]. For the palladium hydride structures, hydrogens were placed in all the octahedral holes of a palladium bulk structure, corresponding to a 1:1 Pd: H ratio, resembling a β-PdH$_x$ structure. The most stable structures were determined with a d-spacing of 2.42 and 2.09 for the 111 and 100 facets, respectively. Activation energies were calculated using the Climbing Image Nudge Elastic Band (CI-NEB) method[88] within the dynamic NEB (DyNEB) implementation[89]. Electronic energies are converted into free energies via a vibrational analysis within the harmonic approximation for adsorbates and an ideal gas approximation for gas phase species, as implemented in the Atomic Simulation Environment (ASE)[80].

The free energy of HCOO$^-_{(aq)}$ was calculated from its equilibrium with HCOOH at the pK$_a$ (3.75), following the relationship $G_{HCOO^-_{(aq)}} = G_{HCOOH_{(aq)}} - \ln(10)k_BT(pH - pK_a)$ [90]. A partial pressure of 5728.86 Pa was applied for the calculation of $G_{HCOOH_{(aq)}}$ from its equilibrium with $G_{HCOOH_{(g)}}$.

A free energy correction of +0.33 eV was added for all molecules including an OCO-backbone, i.e., CO$_{2(g)}$,*H-CO$_2$,*COOH, HCOO$^-$ in order to correct systematic errors of DFT when applying the BEEF-vdW XC functional[91,92].

Noted that, while in the transition state search of *H-CO$_2$, the unit cell explicitly contained CO$_2$ and HCOO$^-$ hovering in the implicit solvent above the electrode surface. However, in Fig. 6, the states CO$_{2(g)}$ and HCOO$^-_{(aq)}$ represent the species in the gas phase and bulk solution, respectively.

## Data availability
The data that support the experimental findings of this study are available from the corresponding author upon reasonable request. The theoretical data and analysis is publicly available on https://github.com/CatTheoryDTU/PdH_formate_vs_CO.

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

## Acknowledgements

Support for this work was provided by the National Research Council of Canada's Materials for Clean Fuels Challenge program, TotalEnergies and McMaster University's Department of Chemical Engineering. All electron microscopy measurements were performed at the Canadian Centre for Electron Microscopy (CCEM) at McMaster University where the LP-TEM holder was provided by a Canadian Foundation for Innovation John R. Evans Leaders Fund (CFI-JELF) grant led by Professor Gianluigi Botton. The computational work was funded by the Villum Foundation through Grant no. 9455. All calculations were performed applying the Niflheim computing cluster at the Technical University of Denmark (DTU).

## Author contributions

A.A. and D.H. conceived the research, A.A. undertook the majority of the experimental work and was responsible for drafting and making revisions to the manuscript. F.I. helped perform ex-situ electrochemical measurements and review the manuscript. O.W.S. and G.K. performed the DFT calculations and theoretical modeling. J.Y. and C.M.A. assisted in conducting the TEM measurements. A. R. performed NMR measurements. L.-A.D. performed SEM imaging. K.E.S. performed ECSA measurements. K.G., N.B., R.B., and L.S. revised the manuscript and provided scientific guidance.

## Competing interests

The authors declare no competing interests.
