## [Peer Review File · Nature Communications]

REVIEWER COMMENTS

Reviewer #1 (Remarks to the Author):

The authors have adequately addressed my concerns from the previous reviews. I only have two more minor comments.

1. The authors should add scale bars to the supplementary movies.
2. I think the paper is now a little too long with regards to the editorial guidelines of Nature Communications (also too many references) and may require some editing. This I leave to the discretion of the journal editors.

Reviewer #2 (Remarks to the Author):

Reviewer report for NCOMMS-23-43158-T

Impact of Palladium/Palladium Hydride Conversion on Electrochemical CO₂ Reduction via In-Situ Transmission Electron Microscopy and Diffraction

By Ahmed M. Abdellah et al.

In this manuscript, the authors provide improvement of previously submitted manuscript. This work primarily leverages on in-situ TEM to monitor Pd based catalyst for electrocatalytic CO₂ reduction (CO₂RR), including formation of PdH_x. The authors also include thorough electrochemical characterisation, and theoretical calculations elucidating the mechanistic pathways towards formate, and CO. There are marked improvement and clarification in terms of data presentation.

However, in terms of insight, I find that most of the findings have been discovered and reported previously in literature. Pd is one of the more attractive CO₂RR catalyst, as it can catalyse formate production at relatively low overpotential. Reaction of CO₂ with H on PdH_x has been studied extensively, and the unusually low overpotential from CO₂ towards formate through CO₂ nucleophilic attack by ads *H (compared to if the reaction goes through bidentate O) has been understood. In terms of fundamental insight, I think not much can be offered by this work.

One strength of this work is the in-situ tracking of the particles during CO₂RR. The findings confirm existing consensus that surface modification will occur under cathodic bias, with significant degree of

nanoparticle sintering after a period of CO₂RR. One possible new aspect that I see from this work is direct observation of hollowed out particles during (after) CO₂RR, but this may not be important in terms of fundamentals understanding of how Pd catalyses CO₂ to formate and/or CO. Undoubtedly, in-situ liquid phase TEM is very hard to do, and the authors have invested considerable careful experiments and expertise in this. However, in my opinion, we are not seeing significant breakthrough in explaining CO₂RR to formate/CO on Pd-based catalyst in this work, other than what is already known in existing literature.

Some specific comments:

1. The authors claimed that the crossover between Formate and CO yield corresponds to shifts towards beta PdHx. In my opinion, the formation of alpha then beta PdHx only can explain formate production at lower overpotential, as higher coverage of highly charged H can be accommodated in beta PdHx, therefore accommodating the nucleophilic attack of CO₂ to form formate, compared to plain Pd.
2. However, the crossover towards CO (g) product is a bit different, separate from PdHx formation. Note that CO (g) is not yet detected before -0.6 V, and the beta PdHx conversion is already complete before -0.3 V. To me, this indicates that the shift towards CO(g) product is not specifically because beta PdHx is formed. I believe this observation makes sense because according to the DFT calculation, the CO₂ → CO pathway only becomes all endergonic with E more cathodic than -0.6 V (cf. Fig 6a). This finding is also in line with the consensus is that CO(g) overtakes as product as *CO adsorbate is slowly building on the surface as the potential becomes more cathodic. In other words, beta PdHx is not a pre-requisite for CO(g) formation, rather it is really because of voltage control that allows *CO sticking to the surface and build coverage (DOI: 10.1038/s41467-021-27793-5; DOI: 10.1021/acscatal.5b00602).
3. CO recovery also has been discussed in DOI: 10.1021/ja511890h, where initially decreased activity of formate production can be recovered once the electrocatalysis is stopped and the catalyst is exposed to fresh electrolyte, possibly clearing out excessive *CO cover.
4. One new aspect could be the observation of hollowed-out particles. This may explain additional resistance of Pd-based catalyst towards deactivation, due to higher availability of surface over time. However, it is not immediately apparent, whether the favourable Pd (111) surface is formed. The effect of hollowed out particles may be reflected in the increasing ECSA (Figure 5e-f), which I pointed out to be counter intuitive to me initially, as there seems to be nanoparticle sintering post cathodic reactions during in-situ LP TEM observation (e.g., Fig 3a-b, Fig4a). Furthermore, we also witnessed growing numbers of particle in in the in-situ TEM substrate. For example, in Figure 4a, with increasing time, we are seeing also growth of (assumedly Pd) particles on the rim (towards the top left). Initially I thought there could be other sources of Pd further out the measurement window. The hollowed-out particles only realised when comparing Fig 4b and Fig4c. Can the authors also confirm that there are similar levels of Pd particle growth and hollowing out on the entire area?
5. The shift of large anodic peaks seems to be weirdly shifted depending on initial E-Hold potential, but with similar peak shape and sizes (Fig 5b). In N₂ saturated electrolyte these features are also present (Figure S16), but the shift is not so bad. Although it is true that some shift in anodic peaks may be expected (attributed to desorbed *H, *CO etc.), but the cathodic peak (around 0.5 in Fig S16) should remain at similar position. (these are also observed in the literature, DOI: 10.1038/s41467-021-27793-5; DOI: 10.1016/j.elecom.2017.05.005. I'm wondering, is there possible pH effect shift? The authors should check this experiment in Fig 5b, but after the same holding time and potential range, do a CV (instead of just forward anodic sweep), so you can see if the cathodic peak around 0.45 V in Figure 5c-5d remains similar (I assume it is the same as the cathodic peak at 0.65V in Figure 5b, in baseline CV)

REVIEWER COMMENTS

Reviewer #1 (Remarks to the Author):

The authors have adequately addressed my concerns from the previous reviews. I only have two more minor comments.

1. The authors should add scale bars to the supplementary movies.
2. I think the paper is now a little too long with regards to the editorial guidelines of Nature Communications (also too many references) and may require some editing. This I leave to the discretion of the journal editors.

Reviewer #2 (Remarks to the Author):

Reviewer report for NCOMMS-23-43158-T
Impact of Palladium/Palladium Hydride Conversion on Electrochemical CO₂ Reduction via In-Situ Transmission Electron Microscopy and Diffraction
By Ahmed M. Abdellah et al.

In this manuscript, the authors provide improvement of previously submitted manuscript. This work primarily leverages on in-situ TEM to monitor Pd based catalyst for electrocatalytic CO₂ reduction (CO₂RR), including formation of PdH_x. The authors also include thorough electrochemical characterisation, and theoretical calculations elucidating the mechanistic pathways towards formate, and CO. There are marked improvement and clarification in terms of data presentation. However, in terms of insight, I find that most of the findings have been discovered and reported previously in literature. Pd is one of the more attractive CO₂RR catalyst, as it can catalyse formate production at relatively low overpotential. Reaction of CO₂ with H on PdH_x has been studied extensively, and the unusually low overpotential from CO₂ towards formate through CO₂ nucleophilic attack by ads *H (compared to if the reaction goes through bidentate O) has been understood. In terms of fundamental insight, I think not much can be offered by this work. One strength of this work is the in-situ tracking of the particles during CO₂RR. The findings confirm existing consensus that surface modification will occur under cathodic bias, with significant degree of nanoparticle sintering after a period of CO₂RR. One possible new aspect that I see from this work is direct observation of hollowed out particles during (after) CO₂RR, but this may not be important in terms of fundamentals understanding of how Pd catalyses CO₂ to formate and/or CO. Undoubtedly, in-situ liquid phase TEM is very hard to do, and the authors have invested considerable careful experiments and expertise in this. However, in my opinion, we are not seeing significant breakthrough in explaining CO₂RR to formate/CO on Pd-based catalyst in this work, other than what is already known in existing literature.

Some specific comments:

1. The authors claimed that the crossover between Formate and CO yield corresponds to shifts towards beta PdH_x. In my opinion, the formation of alpha then beta PdH_x only can explain formate production at lower overpotential, as higher coverage of highly charged H can be accommodated in beta PdH_x, therefore accommodating the nucleophilic attack of CO₂ to form formate, compared to plain Pd.
2. However, the crossover towards CO (g) product is a bit different, separate from PdH_x formation. Note that CO (g) is not yet detected before -0.6 V, and the beta PdH_x conversion is already complete

before -0.3 V. To me, this indicates that the shift towards CO(g) product is not specifically because beta PdHx is formed. I believe this observation makes sense because according to the DFT calculation, the CO₂ → CO pathway only becomes all endergonic with E more cathodic than -0.6 V (cf. Fig 6a). This finding is also in line with the consensus is that CO(g) overtakes as product as *CO adsorbate is slowly building on the surface as the potential becomes more cathodic. In other words, beta PdHx is not a pre-requisite for CO(g) formation, rather it is really because of voltage control that allows *CO sticking to the surface and build coverage (DOI: 10.1038/s41467-021-27793-5; DOI: 10.1021/acscatal.5b00602).

3. CO recovery also has been discussed in DOI: 10.1021/ja511890h, where initially decreased activity of formate production can be recovered once the electrocatalysis is stopped and the catalyst is exposed to fresh electrolyte, possibly clearing out excessive *CO cover.

4. One new aspect could be the observation of hollowed-out particles. This may explain additional resistance of Pd-based catalyst towards deactivation, due to higher availability of surface over time. However, it is not immediately apparent, whether the favourable Pd (111) surface is formed. The effect of hollowed out particles may be reflected in the increasing ECSA (Figure 5e-f), which I pointed out to be counter intuitive to me initially, as there seems to be nanoparticle sintering post cathodic reactions during in-situ LP TEM observation (e.g., Fig 3a-b, Fig4a). Furthermore, we also witnessed growing numbers of particle in in the in-situ TEM substrate. For example, in Figure 4a, with increasing time, we are seeing also growth of (assumedly Pd) particles on the rim (towards the top left). Initially I thought there could be other sources of Pd further out the measurement window. The hollowed-out particles only realised when comparing Fig 4b and Fig4c. Can the authors also confirm that there are similar levels of Pd particle growth and hollowing out on the entire area?

5. The shift of large anodic peaks seems to be weirdly shifted depending on initial E-Hold potential, but with similar peak shape and sizes (Fig 5b). In N₂ saturated electrolyte these features are also present (Figure S16), but the shift is not so bad. Although it is true that some shift in anodic peaks may be expected (attributed to desorbed *H, *CO etc.), but the cathodic peak (around 0.5 in Fig S16) should remain at similar position. (these are also observed in the literature, DOI: 10.1038/s41467-021-27793-5; DOI: 10.1016/j.elecom.2017.05.005. I'm wondering, is there possible pH effect shift? The authors should check this experiment in Fig 5b, but after the same holding time and potential range, do a CV (instead of just forward anodic sweep), so you can see if the cathodic peak around 0.45 V in Figure 5c-5d remains similar (I assume it is the same as the cathodic peak at 0.65V in Figure 5b, in baseline CV)

REVIEWER COMMENTS

Reviewer #1 (Remarks to the Author):

The authors have adequately addressed my concerns from the previous reviews. I only have two more minor comments.

1. The authors should add scale bars to the supplementary movies.
2. I think the paper is now a little too long with regards to the editorial guidelines of Nature Communications (also too many references) and may require some editing. This I leave to the discretion of the journal editors.

Reviewer #2 (Remarks to the Author):

Reviewer report for NCOMMS-23-43158-T

Impact of Palladium/Palladium Hydride Conversion on Electrochemical CO₂ Reduction
via In-Situ Transmission Electron Microscopy and Diffraction

By Ahmed M. Abdellah et al.

In this manuscript, the authors provide improvement of previously submitted manuscript. This work primarily leverages on in-situ TEM to monitor Pd based catalyst for electrocatalytic CO₂ reduction (CO₂RR), including formation of PdH_x. The authors also include thorough electrochemical characterisation, and theoretical calculations elucidating the mechanistic pathways towards formate, and CO. There are marked improvement and clarification in terms of data presentation.

However, in terms of insight, I find that most of the findings have been discovered and reported previously in literature. Pd is one of the more attractive CO₂RR catalyst, as it can catalyse formate production at relatively low overpotential. Reaction of CO₂ with H on PdH_x has been studied extensively, and the unusually low overpotential from CO₂ towards formate through CO₂ nucleophilic attack by adsorbed H (compared to if the reaction goes through bidentate O) has been understood. In terms of fundamental insight, I think not much can be offered by this work.

One strength of this work is the in-situ tracking of the particles during CO₂RR. The findings confirm existing consensus that surface modification will occur under cathodic bias, with significant degree of nanoparticle sintering after a period of CO₂RR. One possible new aspect that I see from this work is direct observation of hollowed out particles during (after) CO₂RR, but this may not be important in terms of fundamentals understanding of how Pd catalyses CO₂ to formate and/or CO. Undoubtedly, in-situ liquid phase TEM is very hard to do, and the authors have invested considerable careful experiments and expertise in this. However, in my opinion, we are not seeing significant breakthrough in explaining CO₂RR to formate/CO on Pd-based catalyst in this work, other than what is already known in existing literature.

Some specific comments:

1. The authors claimed that the crossover between Formate and CO yield corresponds to shifts towards beta PdH_x. In my opinion, the formation of alpha then beta PdH_x only can explain formate production at

lower overpotential, as higher coverage of highly charged H can be accommodated in beta PdHx, therefore accommodating the nucleophilic attack of CO₂ to form formate, compared to plain Pd.

2. However, the crossover towards CO (g) product is a bit different, separate from PdHx formation. Note that CO (g) is not yet detected before -0.6 V, and the beta PdHx conversion is already complete before -0.3 V. To me, this indicates that the shift towards CO(g) product is not specifically because beta PdHx is formed. I believe this observation makes sense because according to the DFT calculation, the CO₂ → CO pathway only becomes all endergonic with E more cathodic than -0.6 V (cf. Fig 6a). This finding is also in line with the consensus is that CO(g) overtakes as product as *CO adsorbate is slowly building on the surface as the potential becomes more cathodic. In other words, beta PdHx is not a pre-requisite for CO(g) formation, rather it is really because of voltage control that allows *CO sticking to the surface and build coverage (DOI: 10.1038/s41467-021-27793-5; DOI: 10.1021/acscatal.5b00602).

3. CO recovery also has been discussed in DOI: 10.1021/ja511890h, where initially decreased activity of formate production can be recovered once the electrocatalysis is stopped and the catalyst is exposed to fresh electrolyte, possibly clearing out excessive *CO cover.

4. One new aspect could be the observation of hollowed-out particles. This may explain additional resistance of Pd-based catalyst towards deactivation, due to higher availability of surface over time. However, it is not immediately apparent, whether the favourable Pd (111) surface is formed. The effect of hollowed out particles may be reflected in the increasing ECSA (Figure 5e-f), which I pointed out to be counter intuitive to me initially, as there seems to be nanoparticle sintering post cathodic reactions during in-situ LP TEM observation (e.g., Fig 3a-b, Fig4a). Furthermore, we also witnessed growing numbers of particle in in the in-situ TEM substrate. For example, in Figure 4a, with increasing time, we are seeing also growth of (assumedly Pd) particles on the rim (towards the top left). Initially I thought there could be other sources of Pd further out the measurement window. The hollowed-out particles only realised when comparing Fig 4b and Fig4c. Can the authors also confirm that there are similar levels of Pd particle growth and hollowing out on the entire area?

5. The shift of large anodic peaks seems to be weirdly shifted depending on initial E-Hold potential, but with similar peak shape and sizes (Fig 5b). In N₂ saturated electrolyte these features are also present (Figure S16), but the shift is not so bad. Although it is true that some shift in anodic peaks may be expected (attributed to desorbed *H, *CO etc.), but the cathodic peak (around 0.5 in Fig S16) should remain at similar position. (these are also observed in the literature, DOI: 10.1038/s41467-021-27793-5; DOI: 10.1016/j.elecom.2017.05.005. I'm wondering, is there possible pH effect shift? The authors should check this experiment in Fig 5b, but after the same holding time and potential range, do a CV (instead of just forward anodic sweep), so you can see if the cathodic peak around 0.45 V in Figure 5c-5d remains similar (I assume it is the same as the cathodic peak at 0.65V in Figure 5b, in baseline CV)

RESPONSE TO REVIEWER COMMENTS

Manuscript ID: NCOMMS-23-43158-T

Title: Impact of Palladium/Palladium Hydride Conversion on Electrochemical CO₂ Reduction via *In-Situ* Transmission Electron Microscopy and Diffraction

Reviewer #1:

The authors have adequately addressed my concerns from the previous reviews. I only have two more minor comments.

Response:

We truly appreciate the comments provided in both the previous round of review and this round of review, and we would like to thank Reviewer #1 for contributing towards improving the overall quality of the work.

1. The authors should add scale bars to the supplementary movies.

Response:

This is an excellent suggestion and we have added scale bars to the supplementary movies.

2. I think the paper is now a little too long with regards to the editorial guidelines of Nature Communications (also too many references) and may require some editing. This I leave to the discretion of the journal editors.

Response:

We agree that this manuscript is longer than a typical Nature Communications paper as it is fairly comprehensive in nature including both experimental measurements (electron microscopy and electrochemistry) alongside computation — all of which being central to the main story. However,

we are hesitant to make substantial changes in terms of moving large portions of the manuscript to the supplementary information as that will result in the final version of the paper looking very different than the version that was peer-reviewed. Furthermore, we are hesitant to remove references as these highlight past relevant research and provide context for the work that is presented in this paper. We would therefore prefer to keep the manuscript at its current length and number of references, however we are open to guidance from the journal editors.

Reviewer #2:

Impact of Palladium/Palladium Hydride Conversion on Electrochemical CO₂ Reduction via In-Situ Transmission Electron Microscopy and Diffraction

By Ahmed M. Abdellah et al.

In this manuscript, the authors provide improvement of previously submitted manuscript. This work primarily leverages on in-situ TEM to monitor Pd based catalyst for electrocatalytic CO₂ reduction (CO₂RR), including formation of PdH_x. The authors also include thorough electrochemical characterisation, and theoretical calculations elucidating the mechanistic pathways towards formate, and CO. There are marked improvement and clarification in terms of data presentation.

Response:

We thank the Reviewer #2 for their encouraging words regarding this work and for the constructive comments that we have addressed through revisions to the manuscript.

However, in terms of insight, I find that most of the findings have been discovered and reported previously in literature. Pd is one of the more attractive CO₂RR catalyst, as it can catalyse formate production at relatively low overpotential. Reaction of CO₂ with H on PdH_x has been studied extensively, and the unusually low overpotential from CO₂ towards formate through CO₂ nucleophilic attack by ads *H (compared to if the reaction goes through bidentate O) has been understood. In terms of fundamental insight, I think not much can be offered by this work.

Response:

We appreciate the reviewer's critical assessment of this work. We acknowledge and understand the presence of PdH_x has been identified and reported previously in literature. However, a key unique aspect of this work is that while some in-situ techniques are valuable for monitoring phase structure transformations in PdH_x/Pd catalysts, they fall short in also observing morphological changes under electrochemical CO₂R conditions which is of fundamental importance for catalyst activity and stability. Furthermore, as referenced by Reviewer #2, formate production at low overpotentials on PdH_x via hydrogenation by electrochemically generated *H has been speculated before [Kanan et al., *JACS*, 137 (2015) 4701]. However, this hypothesis was arrived at by changing the concentrations of reaction relevant species in a three-electrode (two compartment) electrochemical cell and measuring the electro-kinetics of the reaction. While insightful, these measurements were lacking a clear identification of the phase structure(s) present under the conditions explored and did not provide any insight into the reaction energetics or proposed identities of the reaction relevant species/mechanisms.

To address the first point regarding the observation of morphological changes, the following sentence was revised in the introduction section of the manuscript:

“While these *in-situ* synchrotron-based techniques enable monitoring of phase structure transformations in the active PdH_x/Pd materials as a function of electrode potential, they do not provide the opportunity to observe morphological changes in the catalyst particles under CO₂ reduction conditions that have a direct implication on catalytic activity and stability.” Please see Page 4 in the revised manuscript.

To address the second point regarding the fundamental insight into the reaction mechanisms for producing formate on PdH_x, the following was revised in the manuscript:

“Formate production through the reaction of CO₂ with *H has been postulated in the past based on electro-kinetic measurements in a three-electrode (two compartment) electrochemical cell in which the concentrations of reaction relevant species were systematically verified⁹. However, the phase structure(s) and morphology of the Pd/PdH_x catalysts that are present under reaction conditions were not identified through *in situ* characterization, and furthermore the reaction

energetics and identity of intermediate reaction species was not rigorously explored through the combination of *in-situ* catalyst characterization measurements correlated with DFT and micro-kinetic modelling.” Please see page 25 in the revised manuscript.

One strength of this work is the *in-situ* tracking of the particles during CO₂RR. The findings confirm existing consensus that surface modification will occur under cathodic bias, with significant degree of nanoparticle sintering after a period of CO₂RR. One possible new aspect that I see from this work is direct observation of hollowed out particles during (after) CO₂RR, but this may not be important in terms of fundamentals understanding of how Pd catalyses CO₂ to formate and/or CO.

Response:

We appreciate that Reviewer #2 has acknowledged an important aspect of this work is the morphological and micro-structural changes occurring in the Pd/PdH_x electrode that are not observable by conventional *in situ* spectroscopic techniques. We believe that these observations are of importance for the fundamental understanding of the performance of PdH_x catalysts in terms of both activity and stability. To clearly communicate this to readers, we have made the following revisions to the manuscript:

“While these *in-situ* synchrotron-based techniques enable monitoring of phase structure transformations in the active PdH_x/Pd materials as a function of electrode potential, they do not provide the opportunity to observe morphological changes in the catalyst particles under CO₂ reduction conditions that have a direct implication on catalytic activity and stability.” Please see Page 4 in the revised manuscript.

Furthermore, we have supplemented the discussion section to communicate the importance that morphology can have on catalytic activity and to highlight research opportunities that can leverage the findings from this work. The specific revisions are as follows:

“It is also important to note that the heterogeneity of surface structures arising from the morphology of electrocatalysis particles should not be overlooked. Herein we have primarily studied the most thermodynamically stable (111) surface as it is the weakest-binding planar surface termination and therefore least likely to be poisoned, while also having the highest propensity for

Pd-H bond breaking that is central for the production of formate. Thus we postulate that the (111) surface is representative of the structures and electrocatalysis trends observed in this work. Detailed screening of various surface terminations including under-coordinated surfaces consisting steps and defects, or potentially more interestingly, over-coordinated surfaces that weaken interactions with potential poisoning species and reaction intermediates is outside the scope of this study. The scientific community is encouraged to consider and apply the design principles outlined herein in future efforts to design and understand Pd/PdH_x catalysts, especially considering that the morphology of the Pd/PdH_x particles and microstructure of the electrode will evolve under electrochemical CO₂R conditions as observed by the *in situ* LP-TEM measurements.” Please see pages 25-26 in the revised manuscript.

Undoubtedly, in-situ liquid phase TEM is very hard to do, and the authors have invested considerable careful experiments and expertise in this. However, in my opinion, we are not seeing significant breakthrough in explaining CO₂RR to formate/CO on Pd-based catalyst in this work, other than what is already known in existing literature.

Some specific comments:

1. The authors claimed that the crossover between Formate and CO yield corresponds to shifts towards beta PdH_x. In my opinion, the formation of alpha then beta PdH_x only can explain formate production at lower overpotential, as higher coverage of highly charged H can be accommodated in beta PdH_x, therefore accommodating the nucleophilic attack of CO₂ to form formate, compared to plain Pd.

Response:

We appreciate this comment from Reviewer #2, as it is in direct agreement with the conclusions that we have arrived at in this combined experimental-computational work. Indeed, initially, we noticed that that the simultaneous transition from alpha-PdH_x to beta-PdH_x could not rationalize the change in product selectivity from formate towards CO, as intuitively with more *H in the PdH_x one would expect an increase in formate production, whereby the opposite was observed. Given this conundrum, we performed constant-potential DFT simulations, which allowed us to identify the source of the change in product selectivity. Specifically, we found this to be due to the electrode potential dependent changes in the reaction energetics for the CO₂ to formate and CO₂

to CO reaction mechanisms. As the conclusions from this manuscript are in line with what Reviewer #2 has stated, it leads us to believe that we should make some revisions to the manuscript to ensure that this point is communicated to readers as clearly as possible. We have therefore made efforts to clarify the writing by implementing the following revisions:

Abstract:

“A microkinetic model based on the calculated reaction energetics revealed that the intercalation of *H into Pd is essential for formate production. However, the CO₂R selectivity changes from formate at low overpotentials towards CO/H₂ at higher overpotentials is likely not a consequence of morphology or phase structure changes (i.e., increased concentrations *H in PdH_x) but is instead due to electrode potential dependent changes in the reaction energetics for the CO₂ to formate and CO₂ to CO reaction pathways.” Please see page 2 in the revised manuscript.

Introduction:

“The impact of the observed transition from Pd to PdH_x was explored by density functional theory (DFT) calculations. Micro-kinetic analyses, based on the latter, indicate that the production of formate is reliant on the presence of surface bound hydrogen, whose abundance increases with cathodic overpotential. However, the CO₂R selectivity shift results from the varying responses in terms of the reaction energetics to the applied electrode potential of the formate and CO reaction pathways, with the latter benefitting more from increased cathodic overpotentials and not due to the phase structure transformations.” Please see page 5 in the revised manuscript.

Results section:

“Based on the described reaction energetics, a microkinetic model was constructed (**Fig. 6b**). The calculated turnover frequencies (TOF) towards formate outweigh the TOF towards CO at electrode potentials between -0.2 and -0.35 V vs RHE. At more negative potentials, the TOF towards both CO and HCOO⁻ increases, although the increase in TOF for CO is much more drastic. The selectivity for CO (TOF towards CO divided by the sum of the TOFs towards both CO and HCOO⁻) increases as a result of the strong potential response calculated for *COOH as described above. Therefore, this analysis indicates the CO₂R selectivity towards CO should increase at more

negative potentials owing to the electrode potential-dependent energetics of the reaction-relevant species.” Please see page 23 in the revised manuscript.

Discussion section:

“The direct participation of (sub)surface *H in CO₂R towards formate suggests that PdH_x catalysts with an increased *H availability as a reactant (and a reduction in the binding strength of *H) is a desirable catalyst design principle. However, at -0.2 V vs RHE where the highest selectivity towards formate is observed, the catalyst is in a mixed α/β-PdH_x phase. At lower electrode potentials complete conversion to the β-PdH_x phase and increased saturation of the lattice with hydrogen does not coincide with an increase in activity/selectivity towards formate. Instead, the opposite trend is observed. DFT alongside micro-kinetic calculations shows that at more negative electrode potentials, the formation of the *COOH intermediate (for producing *CO) becomes energetically more favourable than the formation of the *H-CO₂ intermediate for producing formate. Therefore, the selectivity change of the CO₂R from producing formate toward CO at more negative electrode potentials is attributed to thermodynamic changes to the energetics of the reaction and not due to the phase transformation leading to increased contents of absorbed H in the PdH_x structures present under reaction conditions.” Please see page 25 in the revised manuscript.

2. However, the crossover towards CO (g) product is a bit different, separate from PdH_x formation. Note that CO (g) is not yet detected before -0.6 V, and the beta PdH_x conversion is already complete before -0.3 V. To me, this indicates that the shift towards CO(g) product is not specifically because beta PdH_x is formed. I believe this observation makes sense because according to the DFT calculation, the CO₂ → CO pathway only becomes all endergonic with E more cathodic than -0.6 V (cf. Fig 6a). This finding is also in line with the consensus is that CO(g) overtakes as product as *CO adsorbate is slowly building on the surface as the potential becomes more cathodic. In other words, beta PdH_x is not a pre-requisite for CO(g) formation, rather it is really because of voltage control that allows *CO sticking to the surface and build coverage (DOI: 10.1038/s41467021-27793-5; DOI: 10.1021/acscatal.5b00602).

Response:

Analogous to our reply to the reviewer's previous comment, we agree that the gradual phase change is likely not responsible for the selectivity transition. Based on our atomistic simulations, combined with thermodynamic and kinetic considerations, we rather identified the potential response of the effective activation free energies varying between the formate and CO reaction mechanisms to be at the hearth of the change in product, with the latter benefitting more from the cathodic overpotential. As noted by Reviewer #2, this change in product preference goes hand in hand with an increase in *COOH and *CO coverage. We do, however, note that according to our simulations on the considered PdH(111) surface, the desorption of CO is generally exergonic, which keeps the *CO coverage well below a complete monolayer at all considered potentials. On other facets the *CO coverage might reach values close to unity, as we highlight based on the PdH(100) facet in Supplementary Fig. 22.

Please refer to our response to the previous comment from Reviewer #2 for details of how we addressed this discussion topic through tangible revisions to the manuscript.

3. CO recovery also has been discussed in DOI: [10.1021/ja511890h](https://doi.org/10.1021/ja511890h), where initially decreased activity of formate production can be recovered once the electrocatalysis is stopped and the catalyst is exposed to fresh electrolyte, possibly clearing out excessive *CO cover.

Response:

We appreciate the Reviewer's acknowledgement that *CO recovery in the context of Pd/PdH_x electrocatalysis for CO₂R has been explored before which we referenced, particularly in ref 9 (DOI: [10.1021/ja511890h](https://doi.org/10.1021/ja511890h)) in the introduction section. Particularly:

“The presence of *CO and *H species (* indicates adsorbed species) on the surface of Pd/PdH_x under CO₂R conditions has been shown to influence the activity, selectivity, and structural evolution of the catalyst^{9, 12, 15}” Please see page 17 in the revised manuscript.

A unique aspect of our work is coupling electrochemical studies with morphological tracking that provides unique insight into degradation mechanisms. Under CO₂R conditions, we observe detachment of some particles from the electrode, the formation of larger particle agglomerates and

a morphological evolution into sponge-like structured particles. Our results lead us to explain that the loss in ECSA is due to this agglomeration/growth, alongside the detachment of some particles that are not re-deposited onto the electrode. Please note that this is the measured ECSA after a surface recovery process in which cyclic voltammetry is applied to remove (strip) the aforementioned adsorbed *CO species. To clearly communicate this aspect to readers, the following revisions have been made to the manuscript:

“This subsequent net decrease in ECSA (observed after *CO removal) is likely due to the detachment of the Pd/PdH_x particles from the electrode surface and some particle agglomeration/growth observed via *in situ* LP-TEM as discussed previously.” Please see page 20 in the revised manuscript.

4. One new aspect could be the observation of hollowed-out particles. This may explain additional resistance of Pd-based catalyst towards deactivation, due to higher availability of surface over time. However, it is not immediately apparent, whether the favourable Pd (111) surface is formed. The effect of hollowed out particles may be reflected in the increasing ECSA (Figure 5e-f), which I pointed out to be counter intuitive to me initially, as there seems to be nanoparticle sintering post cathodic reactions during in-situ LP TEM observation (e.g., Fig 3a-b, Fig4a). Furthermore, we also witnessed growing numbers of particle in in the in-situ TEM substrate. For example, in Figure 4a, with increasing time, we are seeing also growth of (assumedly Pd) particles on the rim (towards the top left). Initially I thought there could be other sources of Pd further out the measurement window. The hollowed-out particles only realised when comparing Fig 4b and Fig4c. Can the authors also confirm that there are similar levels of Pd particle growth and hollowing out on the entire area?

Response:

Thank you for your insightful feedback. There are similar levels of Pd particle growth and hollowing out on the entire area of working electrode as shown in STEM imaging (Supplementary Fig. 12). To clearly communicate these points to readers, we have revised Supplementary Fig. 12 as follows to show STEM images with different magnifications (to show larger regions of the electrode) and at different locations on the electrode:

Supplementary Fig. 12. HAADF-STEM images of *in-situ* TEM working electrode after CO₂ electrolysis. Please see page 17 in the revised Supplementary Information. Furthermore, the following revision was made to the manuscript to address this change:

“This increase was attributed to the introduction of porosity into the Pd/PdH_x particles that occurred over all regions of the electrode as revealed by *ex-situ* HAADF-STEM imaging of the electrodes after CO₂R (Supplementary Fig. 12) as discussed previously.” Please see page 19 in the revised manuscript.

5. The shift of large anodic peaks seems to be weirdly shifted depending on initial E-Hold potential, but with similar peak shape and sizes (Fig 5b). In N₂ saturated electrolyte these features are also present (Figure S16), but the shift is not so bad. Although it is true that some shift in anodic peaks may be expected (attributed to desorbed *H, *CO etc.), but the cathodic peak (around 0.5 in Fig S16) should remain at similar position. (these are also observed in the literature, DOI: 10.1038/s41467-021-27793-5; DOI: 10.1016/j.elecom.2017.05.005. I'm wondering, is there possible pH effect shift? The authors should check this experiment in Fig 5b, but after the same holding time and potential range, do a CV (instead of just forward anodic sweep), so you can see if the cathodic peak around 0.45 V in Figure 5c-5d remains similar (I assume it is the same as the cathodic peak at 0.65V in Figure 5b, in baseline CV

Response:

We fully agree with Reviewer #2 that the difference in pH between the 0.1M KHCO₃ solution saturated with CO₂ and the solution saturated with N₂ can play a role in the resulting cyclic voltammetry behaviour. The pH under CO₂ saturation ~6.8 while under N₂ saturation it is ~8.3 [DOI:10.1039/C4CC03099K], which are values we also measured experimentally with a pH meter. When calibrating and converting our reference electrode measurements from Ag/AgCl we accounted for the ~89 mV difference arising from a pH difference of 1.5 when converting to RHE. To clearly communicate this point to readers, the following sentence was added into the revised manuscript:

“Please note, the conversion between the Ag/AgCl reference electrode used to carry out these measurements and the RHE scale took into account the pH difference between these two experimental conditions (pH of 6.8 for CO₂ purged electrolyte versus 8.3 for N₂ purged).” Please see page 17 in the revised manuscript.

Despite ensuring we did this conversion correctly, the reviewer is correct in that the PdO reduction peak is observed at a slightly more negative electrode potential when the electrolyte is saturated with N₂ (Figure S16) versus CO₂. This shift could likely be due to a pH effect on the reduction of PdO, as reported previously in the literature is likely a pH effect [DOI: org/10.1016/j.apcatb.2014.07.031]. The differ between (Fig 5b) and Figure S16 is due to the different electrolyte pH which could impact the PdO reduction. It is also important to mention that

the change in the cathodic peak location observed in Fig 5b and 5c-5d is due to a change in the scan rate from 20mV/s to 50 mV/s, respectively. A lower scan rate (20mV/s) for Fig. 5b was used to enable us to more clearly distinguish the individual redox peaks that became deconvoluted at higher scan rates (50 mV/s) as shown in the Figure below. Also shown in the Figure below is the impact that a higher scan rate has on the cathodic peak being observed at a more negative electrode potential.

Figure R1. Positive linear sweep voltammetry following a 1 min electrode potential holds at various CO₂R electrode potentials, along with baseline cyclic voltammetry curves collected in CO₂ saturated 0.1 M KHCO₃ electrolyte with a scan rate of **a** 50 mV/sec and **b** 20mV/sec. The purpose of this figure is to show that: (1) At 20mV/s, the individual redox features are easier to distinguish; and (2) A higher cyclic voltammetry potential scan rate leads to the cathodic peak being observed at a lower electrode potential.

To clearly communicate this point regarding scan rate to readers, we have modified the figure caption as shown below.

“Fig. 5 Electrochemical CO₂R selectivity and surface recovery of Pd particles after poisoning by adsorbed *CO species. a Faradaic efficiencies (left y-axis) and partial current densities (right y-axis) for the production of formate, H₂ and CO. **b** Positive linear sweep voltammetry following a 3 min electrode potential hold at various CO₂R electrode potentials, along with baseline cyclic voltammetry curves collected in CO₂ saturated 0.1 M KHCO₃ electrolyte with a scan rate of 20 mV/sec. **c** and **d** Cyclic voltammetry measurements including the 1st and 2nd cycle following

varying durations of an electrode potential hold at -0.1 V vs. RHE at scan rate 50 mV/sec.” Please see pages 20-21 in the revised manuscript.

Regarding the suggested cyclic voltammetry measurements, we have conducted these measurements and found that the applied potential used for chronoamperometry had no impact on the location of the cathodic peak (Figure R2, below). However, we did not include the negative potential sweep in this figure as we felt it would convolute the analysis of the results and detract from the main points of discussion.

Figure R2. Cyclic voltammetry following a 3 min electrode potential hold at various CO₂R electrode potentials, along with baseline cyclic voltammetry curves collected in CO₂ saturated 0.1 M KHCO₃ electrolyte with a scan rate of 20 mV/sec.

While we are not incorporating Figure R2 into the revised version of the manuscript, based on this comment from Reviewer #2 we felt that it would be beneficial to make some revisions to this section of the manuscript to clarify the discussions with the ultimate goal of making this section easier for readers to understand the measurements, and more specifically to illustrate how the

electrochemical data supports the conclusions. Particularly, the following sections have been revised as shown below:

“To investigate the presence of these species, electrode potential holds under CO₂R conditions were carried out on the Pd-decorated large-format electrode followed by cyclic voltammetry measurements to determine the subsequent electrochemical response. Initially, the electrodes were held for varying amounts of time at different electrochemical CO₂R-relevant potentials in CO₂ saturated 0.1M KHCO₃. Without relaxing to open circuit potential, the electrode potential was then swept by linear sweep voltammetry up to 1.2 V vs RHE, which enabled us to observe *CO

stripping peaks to provide insight into *CO surface poisoning (Fig. 5b). Following this sweep, cyclic voltammetry was conducted until a steady state profile was collected, denoted in Fig. 5b as the “baseline CV” measurement. For the linear sweep voltammetry measurements immediately following the chronoamperometric hold under CO₂R conditions (3 min hold at potentials from -0.1 to -0.5 V vs RHE), two oxidation peaks were observed and likely attributed to the oxidation of adsorbed surface *CO species or the desorption/deintercalation of H species⁶²⁻⁶⁷. For example, a 3-minute electrode potential hold at -0.1 V vs RHE led to a subsequent linear

sweep voltammetry measurement with a prominent oxidation feature at electrode potentials < 0.5 V vs RHE, attributed to the desorption/deintercalation of H species. A buildup of adsorbed *CO

species was also indicated by the subtle oxidation peak observed at ca. 0.9 V vs RHE (shown at higher magnification in the inset of Fig. 5b). Applying more negative electrode potentials during the chronoamperometry potential hold, the subsequent linear sweep voltammetry measurements showed that the H desorption/deintercalation peaks shifted to higher potentials, likely arising from the higher concentration of accumulated adsorbed *CO species at more negative electrode potentials as claimed previously⁶⁷, as well as an increased amount of H absorbed into the PdH_x lattice as demonstrated by *in-situ* LP-TEM/SAD measurements.” Please see page 17 in the revised manuscript.

“After a 3-min electrode potential hold at potentials ranging from -0.1 to -0.5 V vs RHE in the N₂ saturated electrolyte, only one oxidation peak at ≤ 0.5 V vs RHE was observed in the subsequent linear sweep voltammetry measurement, attributed to desorption/deintercalation of H from PdH_x. Substantial shifts in the electrode potential of these oxidation features were not observed when

more negative chronoamperometry potentials were applied, providing evidence that the shifts in the H-desorption/deintercalation peaks observed in the case of CO₂ saturated 0.1 M KHCO₃ were largely due to the presence of the adsorbed *CO species and to a lesser extent from the increased concentration of absorbed H in the PdH_x structure.” Please see pages 17-18 in the revised manuscript.

“For electrochemical CO₂R measurements, an increased current density was observed at more negative electrode potentials (Fig. 5a). Over the course of the electrode potential holds used to measure CO₂R activity and selectivity, a decrease in the current density for CO₂R was observed with time (Supplementary Fig. 17), potentially due to gradual poisoning of the Pd/PdH_x surface with *CO.” Please see page 18 in the revised manuscript.

“To gain insight into these simultaneous processes, electrochemically active surface area (ECSA) values were estimated using double-layer capacitance measurements at various stages throughout the course of the chronoamperometry hold and subsequent linear sweep voltammetry measurements detailed in the previous paragraph and shown in Figure 5b. ECSA values were estimated by conducting cyclic voltammetry measurements between 0.2 and 0.4 V vs RHE at varying scan rates as outlined in more detail in Supplementary Note 5 and Fig. 18 of the supplementary information. This route was selected for ECSA estimation as hydrogen underpotential deposition (H_{upd}) measurement could not provide reliable measurements as a significant portion of the current measured in the potential region attributed to H_{upd} for Pd was due to either H adsorption/intercalation or desorption/deintercalation. Moreover, it is important to note that reliable ECSA measurements in the in-situ LP-TEM electrochemical cell were difficult to conduct owing to the small size of the electrodes, which led to very small currents obtained during CV measurements that were significantly impacted by the double-layer capacitance of the underlying glassy carbon electrode. The ECSA for the electrodeposited Pd/PdH_x particles was estimated before ($\text{ECSA}_{t=0}$) and after chronoamperometric potential holds at -0.1 V vs RHE in CO₂-saturated 0.1 M KHCO₃ for durations ranging from 3 to 45 mins ($\text{ECSA}_{t=3 \text{ to } t=45}$). Following ECSA measurements, cyclic voltammetry scans from -0.1 to 1.3 V vs RHE were applied to remove

adsorbed *CO species and restore the “clean” Pd surface. Results of this measurement are shown in Fig. 5c, demonstrating a *CO stripping peak between ca. 0.9 and 1.1 V vs RHE with an increased magnitude of the peak observed with increasing electrode potential hold times. The H desorption/deintercalation are not observed in these cyclic voltammetry measurements as adsorbed/intercalated H species were removed at the electrode potentials applied during the measurements used for ECSA estimation. After the first cycle where *CO species removal was observed (Fig. 5c), subsequent cyclic voltammetry cycles showed negligible differences to each other indicating that the electrode had reached steady state and a pristine Pd surface was recovered. Fig. 5d therefore plots the 2nd cycle as a representative example.” Please see pages 18-19 in the revised manuscript.

“To track the impact of *CO poisoning on the ECSA of the Pd/PdH_x particles during CO₂R, the $ECSA_{t=x}/ECSA_{t=0}$ was estimated (Fig. 5e), where time (t) indicates the duration of the electrode potential hold at -0.1 V vs RHE. When the electrode potential hold period was prolonged from 3 min to 45 mins, the $ECSA_{t=x}/ECSA_{t=0}$ ratio was reduced from 1.01 to 0.80, demonstrating an approximately 20% loss in surface area. This reduction in ECSA could be recovered using cyclic voltammetry to strip *CO and restore the pristine Pd surface, indicating the loss in ECSA observed immediately following the electrode potential hold likely arose due to *CO poisoning. It was then desirable to identify ECSA changes following longer electrode potential holds under CO₂R conditions. 1hr electrode potential holds were therefore conducted sequentially at increasingly more negative electrode potentials, starting at -0.1 V vs RHE and proceeding in increments of 100 mV down to -0.5 V vs RHE. Between each 1hr electrolysis hold, repeated cyclic voltammetry measurements were conducted to clean the Pd surface and ECSA values were measured by double-layer capacitance to calculate the $ECSA/ECSA_{t=0}$ ratios shown in Fig. 5f. The electrolyte was also replaced with fresh electrolyte to remove possible contaminants or liquid phase CO₂R products that could impact subsequent measurements before subsequent electrode potential holds and electrochemical measurements were applied. The calculated $ECSA/ECSA_{t=0}$ after a 1hr electrode potential hold at -0.2 V vs RHE and cyclic voltammetry cleaning showed the highest value of 1.5. This increase was attributed to the introduction of porosity into the Pd/PdH_x particles that occurred over all regions of the electrode as revealed by *ex-situ* HAADF-STEM imaging of the electrodes

after CO₂R (Supplementary Fig. 12) as discussed previously. At more negative electrode potential holds from -0.3 to -0.5 V vs. RHE, the calculated ECSA /ECSA_{t=0} decreased from 1.3 to 0.9, respectively. It is interesting that the normalized ECSA decreases (after surface recovery) at more negative potential, despite the observation of redeposition of smaller Pd particles under cathodic potentials (Fig. 4a). This subsequent net decrease in ECSA (observed after *CO removal) is likely due to the detachment of the Pd/PdH_x particles from the electrode surface and some particle agglomeration/growth observed via *in situ* LP-TEM as discussed previously. Similar particle detachment morphological changes were also observed on the large-format electrodes after a one hour electrode potential hold at -0.5 V vs RHE (Supplementary Fig. 19 and Supplementary Fig. 20), reinforcing the fact that Pd/PdH_x particle detachment was prevalent at these conditions and responsible for the observed ECSA decrease.” Please see pages 19-20 in the revised manuscript.

SUMMARY

Overall, we would like to express our gratitude to both Reviewers for their thoughtful and constructive feedback. These comments have enabled us to make revisions to this manuscript which we hope have significantly improved the quality and clarity of the work.

REVIEWERS' COMMENTS

Reviewer #2 (Remarks to the Author):

Dear Authors,

Thank you for persisting and improving the article. The authors have satisfactorily addressed all my concerns.

One final suggestion is to include the "baseline CV" into the inset of Figure 5b, to highlight the difference in the *CO desorption. The rest of minor corrections can be handled by the editorial team.

RESPONSE TO REVIEWER COMMENTS

Manuscript ID: NCOMMS-23-43158-T

Title: Impact of Palladium/Palladium Hydride Conversion on Electrochemical CO₂ Reduction via *In-Situ* Transmission Electron Microscopy and Diffraction

Reviewer #2:

Dear Authors,

Thank you for persisting and improving the article. The authors have satisfactorily addressed all my concerns.

One final suggestion is to include the "baseline CV" into the inset of Figure 5b, to highlight the difference in the *CO desorption. The rest of minor corrections can be handled by the editorial team.

Response:

This is an excellent suggestion and we have included the "baseline CV" into the inset of Figure 5b as shown below. Please see page 20 in the revised manuscript.